# The power and promise of genetic mapping from *Plasmodium falciparum* crosses utilizing human liver-chimeric mice

Katrina A. Button-Simons [1✉], Sudhir Kumar [2], Nelly Carmago[2], Meseret T. Haile[2], Catherine Jett[3], Lisa A. Checkley[1], Spencer Y. Kennedy [2], Richard S. Pinapati [4], Douglas A. Shoue[1], Marina McDew-White[5], Xue Li[5], François H. Nosten [6,7], Stefan H. Kappe[2], Timothy J. C. Anderson [5], Jeanne Romero-Severson[8], Michael T. Ferdig[1], Scott J. Emrich[9], Ashley M. Vaughan[2] & Ian H. Cheeseman [3✉]

Genetic crosses are most powerful for linkage analysis when progeny numbers are high, parental alleles segregate evenly and numbers of inbred progeny are minimized. We previously developed a novel genetic crossing platform for the human malaria parasite *Plasmodium falciparum*, an obligately sexual, hermaphroditic protozoan, using mice carrying human hepatocytes (the human liver-chimeric FRG NOD huHep mouse) as the vertebrate host. We report on two genetic crosses—(1) an allopatric cross between a laboratory-adapted parasite (NF54) of African origin and a recently patient-derived Asian parasite, and (2) a sympatric cross between two recently patient-derived Asian parasites. We generated 144 unique recombinant clones from the two crosses, doubling the number of unique recombinant progeny generated in the previous 30 years. The allopatric African/Asian cross has minimal levels of inbreeding and extreme segregation distortion, while in the sympatric Asian cross, inbred progeny predominate and parental alleles segregate evenly. Using simulations, we demonstrate that these progeny provide the power to map small-effect mutations and epistatic interactions. The segregation distortion in the allopatric cross slightly erodes power to detect linkage in several genome regions. We greatly increase the power and the precision to map biomedically important traits with these new large progeny panels.

[1] Eck Institute for Global Health, Department of Biological Sciences, University of Notre Dame, Notre Dame, IN, USA. [2] Center for Global Infectious Disease Research, Seattle Children's Research Institute, Seattle, WA, USA. [3] Host Pathogen Interactions Program, Texas Biomedical Research Institute, San Antonio, TX, USA. [4] Nimble Therapeutics, Madison, WI, USA. [5] Disease Intervention and Prevention Program, Texas Biomedical Research Institute, San Antonio, TX, USA. [6] Shoklo Malaria Research Unit, Mahidol-Oxford Tropical Medicine Research Unit, Mahidol University, Mae Sot, Thailand. [7] Centre for Tropical Medicine and Global Health, Nuffield Department of Medicine Research Building, University of Oxford Old Road Campus, Oxford, UK. [8] Department of Biological Sciences, University of Notre Dame, Notre Dame, IN, USA. [9] Univeristy of Tennessee, Knoxville, TN, USA. ✉email: kbuttons@nd.edu; ianc@txbiomed.org

Eukaryotic parasites inflict a high burden of morbidity and mortality particularly in the developing world. Control of these pathogens is threatened by drug resistance[1,2] and understanding the genetic architecture of resistance is essential for the design of further interventions. Previous studies in *Plasmodium*, *Trypanosome,* and *Leishmania* parasites revealed that genetic architecture of drug resistance is often complex[3–6]. For example, emergent artemisinin resistance in the human malaria parasite, *Plasmodium falciparum*, has been causally associated with multiple independent mutations in one gene, *kelch13* (*pfk13*), which explain nearly all the variation in this phenotype[7–9]. However, mutations in the *ferredoxin* (*pffd*), *apicoplast ribosomal protein 10* (*pfarps10*), *multi-drug resistance protein 2* (*pfmdr2*), and *chloroquine resistance transporter* (*pfcrt*) genes are significantly associated with artemisinin resistance, and have been proposed to constitute a genetic background highly predisposed to the development of resistance[7]. Several techniques have been used to identify the genetic determinants of complex phenotypes in eukaryotic pathogens including genome-wide association studies (GWAS)[7,10], in vitro selections[8], quantitative trait loci (QTL) analysis in controlled genetic crosses[11–14] and bulk segregant analysis/linkage group selection/extreme QTL (BSA/LGS/XQTL)[5,15]. Controlled genetic crosses offer a powerful way to dissect the genetic architecture of complex traits. For example, the progeny of an experimental cross revealed that *P. falciparum* sensitivity to quinine was associated with loci on chromosomes 5, 7, and 13, with the chromosome 5 and 7 loci containing known drug resistance transporters *pfcrt* and *pfmdr1*[3].

*P. falciparum* has the potential to be a particularly powerful genetic mapping system due to its unusually high recombination rate (11–13.3 kb/cM[13,16,17]), haploid state for most of the life cycle, and the ability to clone blood stage progeny in vitro, creating effectively immortal $F_1$ mapping populations in a single generation. In addition, *P. falciparum* has a small genome (23 Mb) and a high quality reference assembly[18] with frequent annotation updates;[19,20] consequently, re-sequencing and comprehensive analysis of the genome of progeny is simple and cost effective[21]. Generating controlled genetic crosses in *P. falciparum*, however, has historically been a difficult and time-consuming process requiring splenectomized chimpanzees in place of a human host. This resulted in only four genetic crosses being performed over a 30-year period. $F_1$ mapping populations from all four previous *P. falciparum* genetic crosses have been small, containing 33, 35, 15, and 27 individual recombinant progeny[13,21]. When compared to the thousands of progeny possible in many plants and fruit flies[22], these numbers are small. To use genetic mapping to elucidate the genetic architecture of emerging drug resistance in *P. falciparum* we need to be able to rapidly create genetic crosses with large numbers of progeny from recent field isolated parasites which exemplify relevant clinical traits such as drug resistance.

Here we report on the production of large numbers of unique recombinant progeny utilizing human liver-chimeric FRG huHep mice infused with human red blood cells. Although these mice were previously reported as an option for *P. falciparum* genetic crosses[23], until now they have failed to produce more progeny than historic crosses. Here we report on two genetic crosses that were carried out using recent clinically derived *P. falciparum* isolates with emerging drug resistance phenotypes. This effort was aided by a progeny characterization bioinformatics framework that filters single nucleotide polymorphisms (SNP) and identifies clonal unique recombinant progeny. We generated genetic maps for each cross (84 and 60 unique recombinant progeny) and provide detailed investigation of inbreeding, plastid inheritance, and cross-over rates. Through simulation and mapping we show that we are better powered to detect genetic associations than

previous crosses, even for small effect sizes. We show that while segregation distortion (SD) can locally reduce power we are still able to detect major effect loci in our expanded progeny panels.

## Results

**Rapid generation of genetic crosses**. During a single year we conducted two independent genetic crosses using multiple FRG NOD huHep mice. The first, an allopatric cross between a laboratory-adapted African line (NF54) and a newly cloned clinical isolate (NHP4026) from the Thai–Myanmar border; the second, a sympatric cross between two newly cloned clinical isolates (MKK2835 and NHP1337) from the Thai–Myanmar border. These crosses yielded 84 and 60 clonal unique recombinant progeny lines respectively. The pipeline to generate recombinant progeny is technically challenging and takes ~6 months (Fig. 1). Initially, we confirmed that the parental lines produced infectious gametocytes that gave rise to infectious sporozoites. After this confirmation, the steps to complete a genetic cross include asexual culture and expansion, gametocyte maturation, mixing of parental gametocytes and transmission to mosquitoes, confirmation of successful mosquito stage development, salivary gland sporozoite isolation and infection of human hepatocytes in the FRG NOD huHep mouse, liver stage development, infusion of human red blood cells, the in vivo transition from liver stage-to-blood stage, the subsequent transition to in vitro blood stage culture coupled with cloning by limiting dilution and finally clonal expansion, confirmation of clonality and genome sequencing of recombinant progeny (Fig. 1).

In total, we initiated three independent crosses (NF54HT-GFP×NHP4026[23], NF54WT×NHP4026, and MKK2835×NHP1337[24]). NF54 and NF54HT-GFP–luc[25] are largely drug sensitive parasite lines that readily produce gametocytes; MKK2835 is an isolate from Thailand that was collected in 2003 before the emergence of artemisinin resistance[26]. NHP4026[26], and NHP1337[26] are recent isolates from the Thai/Myanmar border with reduced in vivo sensitivity to artemisinin (Table 1). The NF54WT×NHP4026 cross was performed to determine if the GFP–luc transgene[25] integration into the genome was the cause of a segregation distortion peak (described below). The progeny from the first crosses were combined (subsequently referred to as NF54×NHP4026) to form one genetic map (described below). A bulk analysis of the uncloned progeny from the MKK2835×NHP1337 cross was previously published[24].

We staged each genetic cross by infecting multiple cages of mosquitoes with mixed gametocyte cultures of our parental lines (Supplemental Table 1). Details of the NF54HT-GFP–luc×NHP4026 cross were previously published[23]. For NF54WT×NHP4026, sporozoites from three individual cages of mosquitoes were used to infect three individual mice by intravenous (IV) injection or mosquito bite (MB) (one cage per mouse). One mouse was infected by IV injection of one million sporozoites dissected from 250 mosquitos with infection prevalence of 73% and median 2 oocysts/infected mosquito. Two mice were infected by MB using cages with 250 mosquitos with prevalence of 73% and 58% and median 3 oocyst/infected mosquito. Assuming no attrition in parasite genotypes, equal initial gametocyte ratios and random mating, this translates to 730 unique recombinant progeny for the IV infection (Eq. 1).

$$250 \text{ mosquitos} \times 0.73 \text{ infection prevalence} \times 0.5 \text{ outcrossing rate} \times$$
$$2 \text{ oocysts per infected mosquito} \times 4 \text{ recombinants per oocyst} = 730 \text{ recombinants}$$

$$\tag{1}$$

Similarly, the MB infections contained up to 1095 and 870 unique recombinants. For MKK2835×NHP1337 four cages of mosquitos were infected with pools of MKK2835 and NHP1337

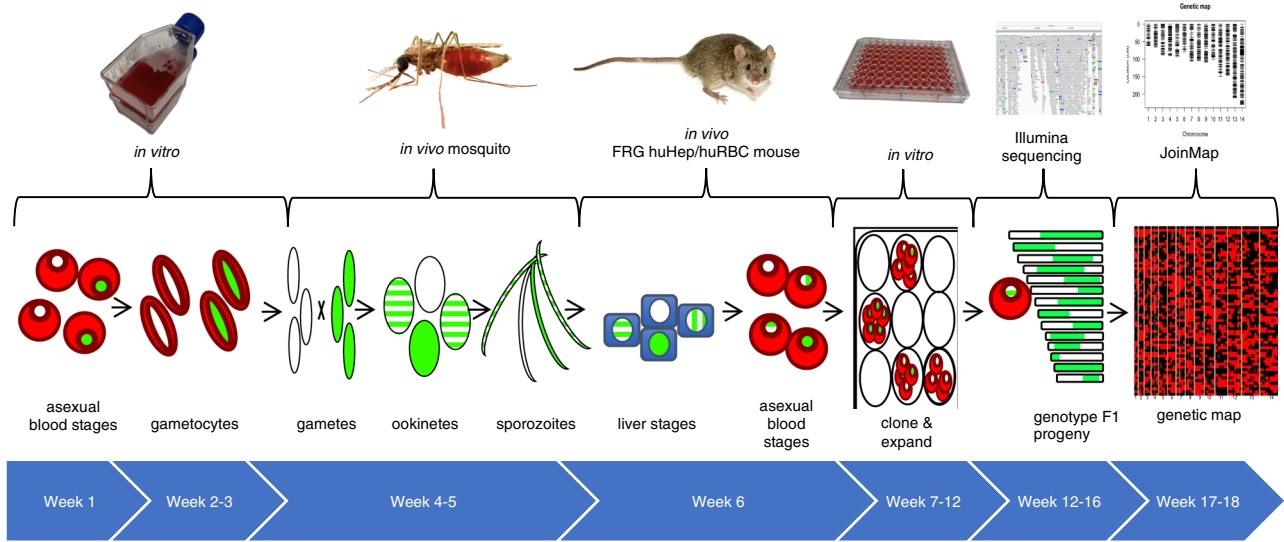

**Fig. 1 Timeline for performing *P. falciparum* crosses in FRG NOD huHep/huRBC mice.** Uncloned F₁ progeny from *P. falciparum* genetic crosses of recent field isolates were recovered within 6 weeks from the start of the asexual stage culture of the two parent lines. Cloning of potential F₁ recombinant progeny takes an additional 6 weeks and next-generation sequencing of potential recombinant progeny and identification of unique recombinants via our pipeline takes an additional 6 weeks. This figure was adapted from Fig. 1a of Vaughan et al.[23]. The mosquito image was adapted from a photo created by James Gathany and the mouse image was created by George Shuklin both were made available for public use through creative commons lisences.

**Table 1 Drug resistance related genotype information and drug phenotype information for parents.**

| Genotype/Phenotype | | Parental parasite line | | | |
|---|---|---|---|---|---|
| Genotype | Drug resistance associations | NF54/NF54GFPLuc[25] | MKK2835 | NHP4026 | NHP1337 |
| *pfdhfr* (N51I) | Pyrimethamine | N | I | I | I |
| *pfdhfr* (C59R) | Pyrimethamine | C | R | R | R |
| *pfdhfr* (S108N) | Pyrimethamine | S | N | N | N |
| *pfgch1* copy number | Pyrimethamine | 5 | 5 | 3 | 4 |
| *pfdhps* (A437G) | Sulfadoxine | G | G | G | G |
| *pfdhps* (K540E) | Sulfadoxine | K | E | E | E |
| *pfmdr* copy number | Quinine, Amodiaquine, Mefloquine, Halofantrine, Lumefantrine | 1 | 1 | 1 | 1 |
| *plasmepsin II/III* copy number | Piperaquine | 1 | 1 | 1 | 1 |
| *pfcrt* (K76T) | Chloroquine | K | T | T | T |
| *pfcrt* (I356T) | Artemisinin derivatives | I | T | T | T |
| *pfk13* (C580Y) | Artemisinin derivatives | C | C | C | Y |
| *pffd* (D193Y) | Artemisinin derivatives | D | Y | Y | Y |
| *pfarps10* (V127M) | Artemisinin derivatives | V | V | M | M |
| *pfmdr2* (T484I) | Artemisinin derivatives | T | I | I | I |
| Phenotype | | NF54/NF54GFPLuc[25] | MKK2835 | NHP4026 | NHP1337 |
| $PC_{1/2}$ | | NA | NA | 8.37 | 7.84 |
| eRRSA | | 47.0 ± 17.5 | 59.1 ± 20.9 | 48.2 ± 8.9 | 14.8 ± 17.4 |
| Chloroquine $IC_{50}$ | | 24.0 ± 6.9 | 514.5 ± 140.1 | 667.3 ± 27.0 | 420.1 ± 41.3 |
| Piperaquine $IC_{50}$ | | 9.9 ± 3.3 | 7.3 ± 2.8 | 8.6 ± 0.3 | 9.2 ± 2.7 |
| Dihydroartemisinin $IC_{50}$ | | 0.9 ± 0.2 | 0.6 ± 0.04 | 1.0 ± 0.5 | 0.5 ± 0.1 |

$PC_{1/2}$ is the patient clearance half-life for parasites treated with artemisinin, a $PC_{1/2}$ of over 5 h defines slow clearance a hall-mark of artemisinin resistance. eRRSA is an improved version of the ring survival assay (RSA)[56] a eRRSA value of less than 30 corresponds to a $PC_{1/2}$ of greater than 5. None of the pfcrt mutation described in Ross et al.[59] were present in any of these parasites. NHP4026 does not contain any other coding mutations in k13.

gametocytes and the cage with the best infections (80% prevalence and median 3 oocysts/infected mosquito, 204 mosquitos) was used to infect a single mouse via IV injection with 2.7 million sporozoites, containing up to 979 unique recombinants. Following exsanguination, the parasitemia of the NF54WT × NHP4026 cross mice were 0.017% (MB), 0.02% (MB) and 0.013% (IV) (Supplemental Table 1). The parasitemia of the MKK2835 × NHP1337 cross mouse was 4.5% (Supplemental Table 1).

**Numbers of unique recombinant progeny.** The F₁ progeny in the blood of the infected FRG NOD huHep/huRBC mouse must be cloned by limiting dilution after exsanguination. Isolated progeny may be non-clonal because a small subset of wells were initiated with more than 1 parasite per well. Additionally, the same recombinant progeny can be cloned more than once because they undergo clonal expansion[27] before cloning. Furthermore, since parents in both crosses produce fertile male and female gametocytes, selfed progeny can also arise. We developed a

bioinformatics pipeline to identify unique clonal recombinant progeny, filtering out non-clonal progeny, selfed progeny, and repeat sampling of the same genotype (see "Methods").

Genetic characterization of previous crosses was initially carried out using restriction fragment length polymorphism (RFLP) or microsatellite (MS) markers[16,28] and unique recombinant progeny were recently sequenced to create a community resource[21]. We initially performed MS genotyping on the progeny from the first two cloning attempts for the NF54 × NHP4026 cross, filtering out most non-clonal, selfed and non-unique recombinant progeny and then performed genome sequencing of select progeny. For subsequent cloning attempts for the NF54 × NHP4026 cross and the MKK2835 × NHP1337 cross, we performed genome sequencing of all cloned parasites. For each progeny, sequencing reads were mapped to the *P. falciparum* genome (version 3)[29], SNPs were called jointly across parents and progeny, and filtered to contain SNPs in the 20.8 Mb core genome[21].

For NF54 × NHP4026, 10,472 high quality bi-allelic SNPs (1 SNP per 2.0 kb) differentiate the two parents. We cloned 175 parasites and sequenced 161 parasites following MS filtering. In total, filtering removed seven progeny with low genome coverage, 25 non-clonal and three selfed progeny (Fig. 2), ultimately resulting in the identification of 140 recombinants, 84 of which were unique (Fig. 2). For MKK2835 × NHP1337, we identified 7198 high quality bi-allelic SNPs (1 SNP per 2.9 kb) that differentiate the two parents. 266 parasites were cloned and sequenced. Filtering removed 18 samples with low coverage, 36 non-clonal, and 149 selfed progeny and resulted in the identification of 63 recombinant progeny (Fig. 2), 60 of which were unique. To maximize the capture of unique recombinant

progeny from each cross we cloned on up to three occasions (hereafter referred to as cloning rounds) from cultures of bulk uncloned progeny or cryopreserved stocks of bulk uncloned progeny frozen just after exsanguination (Supplemental Table 2). Across all crosses, each cloning round produced nearly unique sets of recombinant progeny, with only one repeat genotype across cloning rounds (Fig. 2 and Supplemental Fig. 1).

**Inbreeding, outbreeding, and plastid inheritance.** We identified stark differences in patterns of outcrossing between the two crosses. NF54 × NHP4026 cloned progeny contained only three selfed NF54 progeny and no selfed NHP4026 progeny (2.1% of all clonal progeny, Fig. 2). In contrast, MKK2835 × NHP1337 cloned progeny contained 144 selfed NHP1337 progeny and five MKK2835 selfed progeny (65% of all clonal progeny, Fig. 2). In both crosses, when cloning was initiated within 5 days of establishing in vitro culture, almost all recombinants were unique (Supplemental Table 2 and Supplemental Fig. 1). The percentage of recombinants that were unique was high whether cloning was initiated from continuous in vitro culture or cryopreserved culture (87–100% for continuous in vitro culture vs. 93% from a thawed cryopreserved bulk culture). However, when cloning was initiated after 14 days of in vitro culture from cryopreserved bulk culture, only 34% and 45% of recombinants were unique (Supplemental Table 2 and Supplemental Fig. 1), likely due to the preferential expansion of a subset of progeny.

*P. falciparum* parasites contain two plastid organelles, the mitochondria and apicoplast, both of which are maternally inherited[30]. Despite *P. falciparum* being hermaphroditic, in previous genetic crosses nearly all plastid genomes in the progeny originated from a single parent[31,32]. In our crosses we observed

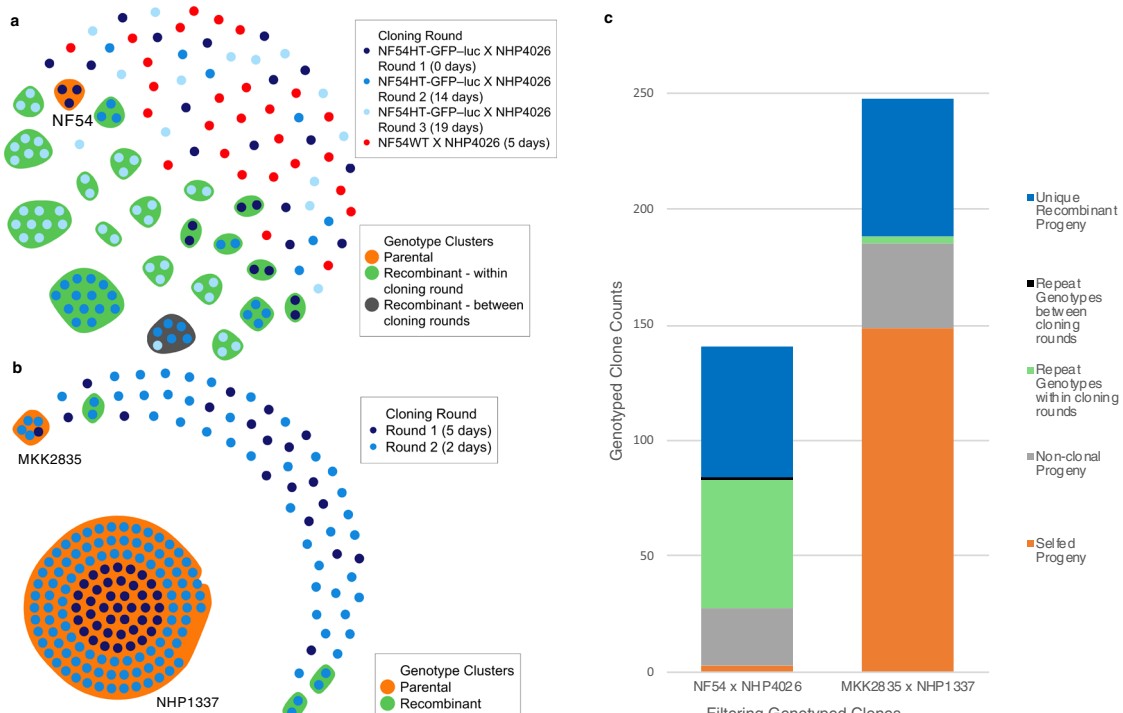

**Fig. 2 Cloning results and recombinant progeny for each cross. a, b** Genotyping results for each cross. Circles represent clonal genotyped progeny. Colored ellipses surround individual cloned progeny of the same genotype, with orange ellipses denoting parental progeny. **a** The cloned progeny from the NF54 × NHP4026 contained three selfed NF54 progeny (orange ellipse). The majority of repeat sampling of the same genotype (green and gray ellipse) occurred in cloning round 3, the gray ellipse denotes the only observed repeat sampling event between cloning rounds (rounds 2 and 3, gray ellipse). **b** The MKK2835 × NHP1337 cross produced 144 selfed NHP1337 progeny, five MKK2835 selfed progeny and few instances of repeat sampling (green ellipses). **c** Progeny were characterized to identify unique recombinant progeny (blue), selfed progeny (orange), non-clonal progeny (gray) and repeat sampling of the same genotype within a cloning round (green) and between cloning rounds (black).

both parental plastid genotypes among the unique recombinant progeny. In the NF54 × NHP4026 cross, 18% of the unique recombinant progeny inherited their plastid genomes from NF54 and in the MKK2835 × NHP1337 cross, 42% of the unique recombinant progeny inherited their plastid genomes from MKK2835.

**Genetic maps and recombination rates**. For each cross, we generated a genetic map (Supplemental Data 3 and Supplemental Data 4) from phased genotype data for all unique recombinant progeny (see Methods). The map length and recombination rate for both crosses are consistent with those reported for previous crosses[16,17,33] (1507 cM and 13.6 kb/cM for NF54 × NHP4026;

1681 cM and 12.1 kb/cM for MKK2835 × NHP1337, Table 2). For the NF54 × NHP4026 genetic map, markers initially sorted into 13 linkage groups rather than the expected 14, with one linkage group containing markers for chromosomes 7 and 14. We identified extreme deviation from expected Mendelian inheritance (see below) which likely explained this result. Adjusting JoinMap parameters (see Methods) separated the 13th linkage group into two groups, recovering distinct sets for chromosomes 7 and 14. For the MKK2835 × NHP1337 cross, all markers separated into 14 linkage groups, corresponding to the known chromosomes.

**Repeatability of segregation distortion**. We phased 5 kb windows of the core genome to indicate inheritance blocks for each unique recombinant progeny (Fig. 3a, b). We observed regions with significant segregation distortion (SD) on chromosomes 7, 12, 13, and 14 in the NF54 × NHP4026 cross ($\chi^2$ test for deviation from expected Mendelian ratio of 1:1, $p < 0.001$). Patterns of allele frequency variation in both replicates of the NF54 × NHP4026 were highly repeatable (Figs. 3a, and 4, concordance correlation coefficient of 0.66). We observed no significant SD in the MKK2835 × NHP1337 recombinant progeny in either cloning round (Fig. 3b and Supplemental Fig. 2). This was consistent with measured bulk allele frequencies after accounting for baseline shifts in genome-wide allele frequency due to selfed progeny on days 2 and 5 after cloning was initiated[24].

The distorted regions on chromosomes 7 and 13 were present in all cloning rounds of the NF54 × NHP4026 cross, including when cloning was initiated immediately after mouse exsanguination (Supplemental Fig. 2). However, the distorted regions on chromosomes 12 and 14 were present only in the NF54HT-

| Cross Progeny number Recombination rate | F₁ Genetic map length | | |
|---|---|---|---|
| HB3 × Dd2 | 35[16] | 1556[16] | 12.1 kb/cM[16] |
| 3D7 × HB3 | 15[21] | | 11.0 kb/cM[33] |
| 7G8 × GB4 | 32[17] | 1655[17] | 12.8 kb/cM[17] |
| GB4 × 803 | 27[13] | | 13.3 kb/cM |
| NF54 × NHP4026 | 84 | 1507 | 13.6 kb/cM |
| MKK2835 × NHP1337 | 60 | 1681 | 12.1 kb/cM |

**Table 2 Number of progeny, genetic map length and recombination rate for historic *P. falciparum* genetic crosses, the NF54 × NHP4026 cross and the MKK2835 × NHP1337 cross.**

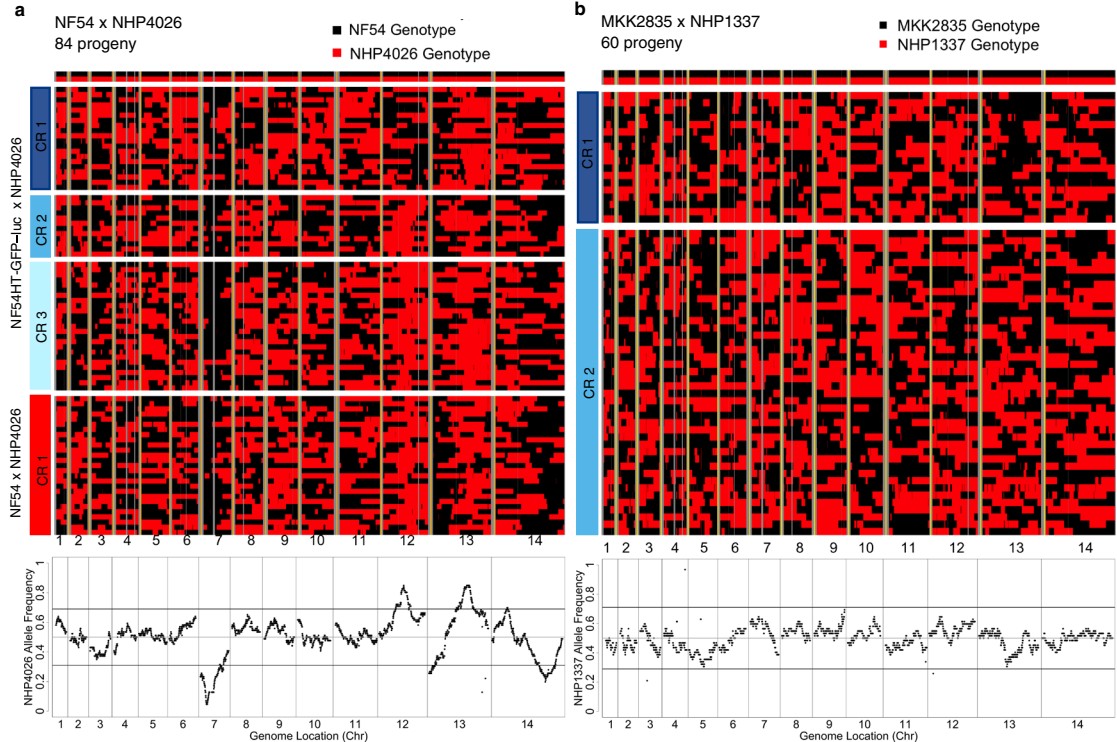

**Fig. 3 Physical maps of recombinant progeny from two genetic crosses.** Physical maps (**a**, **b**) depict inheritance patterns in 5 kb blocks across the core genome[21] (x axis) for each progeny (y axis). Non-core regions of the genome with no variant calls are shown in gray and yellow shows chromosome boundaries. Each heatmap is broken into sections by cloning round. The lower panels show allele frequencies across the genome for the unique recombinant progeny and the horizontal black lines show a cut-off for segregation distortion, a deviation from the expected ratio of 1:1 at $p = 0.001$. **a** The physical map for the NF54 × NHP4026 progeny shows regions where haplotype blocks deviate significantly from the expected 1:1 ratio. **b** The physical map for MKK2835 × NHP1337 shows even inheritance ratios across the genome, in line with Mendelian expectations.

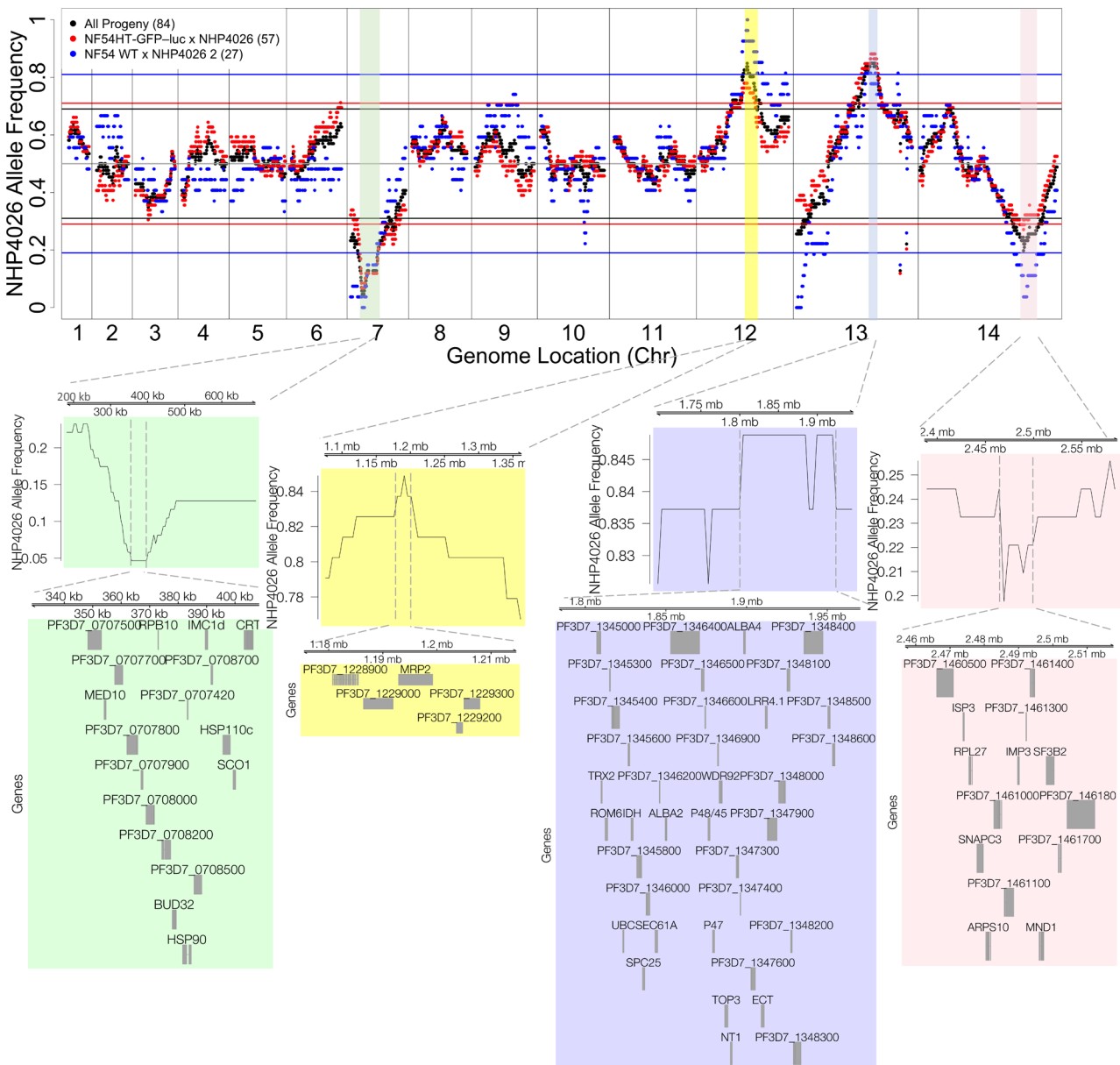

**Fig. 4 Replicated segregation distortion in the NF54 × NHP4026 cross.** Frequency of the NHP4026 SNP alleles in unique recombinant progeny in NF54 × NHP4026 is highly repeatable across biological replicates (black—all progeny, red—progeny from biological replicate 1, blue—progeny from biological replicate 2). Horizontal lines represent significance thresholds ($\chi^2$ test $p = 0.001$) for segregation distortion for each corresponding set of progeny. Colored regions (green, yellow, lilac, pink) show significant segregation distortion in both biological replicates. Genes are shown for the peak regions of segregation distortion.

GFP–luc × NHP4026 cross cloning rounds that were initiated at least 5 days after mouse exsanguination (Supplemental Fig. S2). The SD on chromosome 13 coincides with the *pf47* locus where the GFP–luc transgene was integrated. We were thus concerned that the integration led to the distortion and repeated the NF54 × NHP4026 cross using the unedited parental NF54. The segregation distortion persisted in the cloned progeny from this updated cross, suggesting that the GFP–luc transgene is not the driver of this distortion and allowing us to combine the progeny from NF54 × NHP4026 in estimating genetic maps.

**Distorted loci**. We examined each distorted locus in the NF54 × NHP4026 cross for plausible driver genes (Fig. 4). Supplemental Data 3 lists the genes and SNPs in each region. On chromosome

7, a 520 kb region containing 121 genes showed significant SD in both biological replicates of the cross ($\chi^2$ test, $p < 0.001$, Fig. 4, Supplemental Data 3) for all cloning rounds (Supplemental Fig. 2). At the SD peak only three progeny inherit the NHP4026 allele. The region with the most extreme SD contains 17 genes (Fig. 4) including *pfcrt* (PF3D7_0709000), mutations in which lead to chloroquine resistance and a fitness disadvantage in some genetic backgrounds[34].

On chromosome 12, a 295 kb region with 71 genes showed SD with an overabundance of NHP4026 alleles when cloning was initiated at least 5 days after exsanguination. The segregation peak contains five genes (Fig. 4) including *pfmrp2* (PF3D7_1229100). SNPs and microindels in *pfmrp2* have significant associations with in vitro response to chloroquine, mefloquine, and piperaquine and in vivo parasite clearance[35].

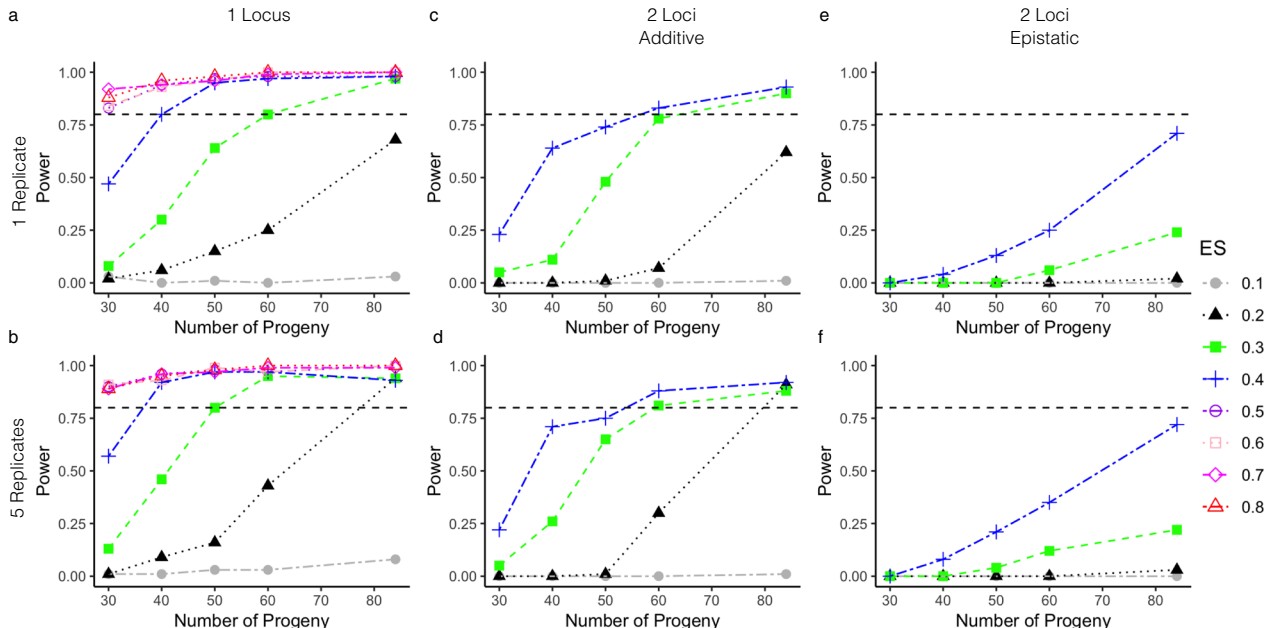

**Fig. 5 Power analysis for different size progeny sets.** Power curves are shown from simulated phenotypes for the NF54 × NHP4026 progeny for different size progeny sets. The top row **a**, **c**, **e** shows power curves where the phenotype only has a single replicate per progeny strain and the bottom row **b**, **d**, **f** shows results for five replicate phenotype values per progeny strain. The first column **a**, **b** shows results for a single locus effect, the second column **c**, **d** shows results for an additive two loci effect and the third column **e**, **f** shows results for an epistatic interaction between two loci. The horizontal black dotted line denotes 80% power.

On chromosome 13, a 230 kb region predominantly inherited from NHP4026 with 56 genes shows SD in all cloning rounds. The SD peak contains 37 genes including *pf47* (PF3D7_1346800) and *pf48/45* (PF3D7_1346700). In the NF54HT-GFP–luc × NHP4026 replicate of this cross the GFP–luc transgene replaces the functional *pf47* locus[23] but the same distortion was also seen when wildtype NF54 was used for the cross, suggesting that GFP–luc transgene is not the driver of the skewed inheritance.

On chromosome 14, a 205 kb region containing 62 genes predominantly inherited from NF54 showed SD when cloning was initiated at least 5 days after exsanguination. The SD peak contains 15 genes including *pfarps10* (PF3D7_1460900) which has been associated with slow clearance of parasites from patient blood after artemisinin combination therapy (ACT) treatment in GWAS studies[7] and is hypothesized to contribute to a permissive background for evolution of *pfk13* mutations which have been shown to confer artemisinin resistance in vitro[9].

**Increased mapping power in an expanded genetic cross.** Small sample size (Table 2) and SD (Supplemental Fig. 3) have limited the power of previous crosses to detect mutations with small effect size (ES), measured as variation attributable to a locus/total variation. We quantified how our expanded progeny sets will improve genetic mapping using simulations. Specifically, we quantified the impact of phenotypic replication, progeny number, and the number of loci on statistical power and mapping resolution using the 84 NF54 × NHP4026 progeny (Fig. 5 and Supplemental Fig. 4). Using progeny panels comparable to previous genetic crosses (*n* = 30–40) only very large ES > 0.5) can be mapped with high power (>80%) (Fig. 5a, b). In contrast, 84 progeny enable mapping of much smaller effect loci (ES = 0.2) at 80% power (Fig. 5a, b). Increasing the number of progeny also increases the locus resolution (Supplemental Fig. 4). At an ES of 0.5, with *n* = 30, we can map to a region containing 58 genes; moreover, with *n* = 84, we can map to a region containing only 17 genes (Supplemental Fig. 4). At small ES we observe similar

large increases in mapping resolution as we increase the size of the progeny set (Supplemental Fig. 4).

Phenotypes are frequently complex in nature[3,4,36,37]. For a trait controlled by two additive loci (non-interacting) that contribute equally to the phenotype, 60 progeny, with replicated phenotypes can detect a significant statistical association at ES = 0.3, whereas 84 progeny are needed to detect a significant statistical association at ES = 0.2 (Fig. 5c). When a trait is controlled by two epistatically interacting loci, 84 progeny with replicated phenotypes provide 75% power to detect a significant statistical association and interaction with ES = 0.4 (Fig. 5e). Replicated phenotypes allow the same power to be achieved with fewer progeny for ES ≥ 0.3 and allow for a trait with ES = 0.2 to be detected for *n* = 84 progeny for additive loci (Fig. 5d). This analysis indicates that the four previous *P. falciparum* crosses (15–35 progeny, conducted in chimpanzee hosts) were underpowered. In progeny sets of this size, reliable detection of significant statistical associations is possible only for phenotypes with large ES (≥0.5) (Fig. 5a, b) and detection of even strong epistatic interaction (Fig. 5e, f) is not possible. Our two new crosses (*n* = 60 and 84 progeny) have higher power and are capable of reliably detecting significant associations for phenotypes with ES as low as 0.2.

Polygenic traits often have unequal contributions from multiple loci. In malaria parasites, several drug resistance phenotypes have known secondary genes that further modify resistance levels[3,4,36,37]. Analysis of our progeny genotypes and simulated phenotype data (Fig. 6) shows that with 84 progeny we can detect secondary loci with ES as low as 0.2 and 0.15. However, with 35 progeny we cannot detect these secondary loci.

**Segregation distortion decreases the resolution and power of mapping.** SD is abundant across all *P. falciparum* genetic crosses generated to date, except our newly generated MKK2835 × NHP1337 cross. We examined the impact of SD on the power to identify causal variants. SD decreases power to detect effects near

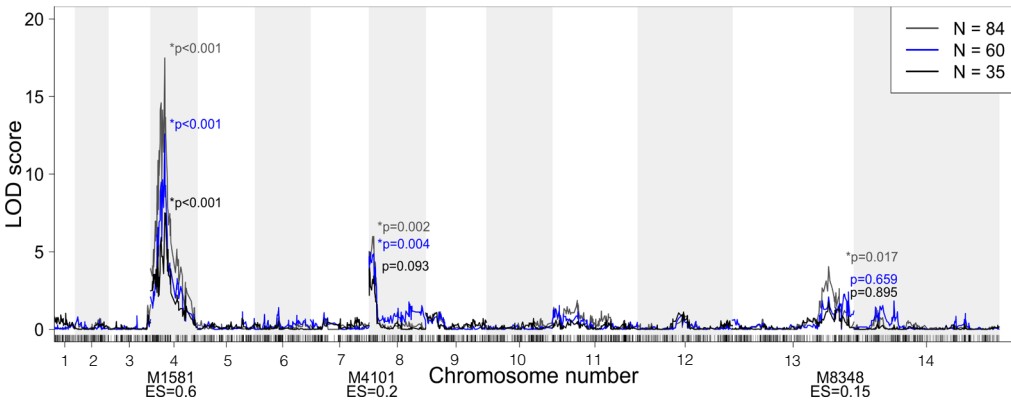

**Fig. 6 Detecting complex associations.** QTL scans of simulated phenotypes with one major (ES = 0.6) and two minor (ES = 0.2 and 0.15) contributing loci for $N$ = 84 (gray), 60 (blue) and 35 progeny (black). The major locus is detected for all sizes of $N$, but only one minor locus is detected for $N$ = 60 progeny and neither minor locus is detected at $N$ = 35 progeny.

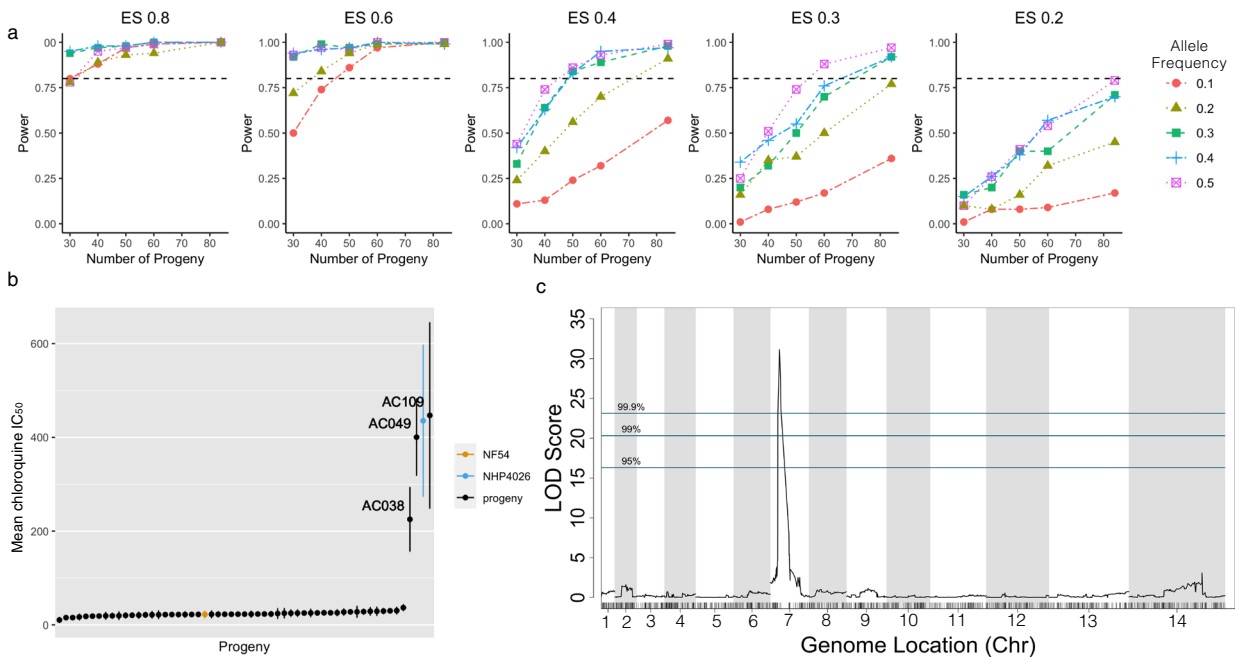

**Fig. 7 Power loss due to segregation distortion. a** Effect of segregation distortion on mapping power with NF54 × NHP4026 progeny with simulated phenotype data at different ES. Each sub-panel shows the relationship between allele frequency and power for different numbers of progeny at a fixed ES. For a high ES, allele frequency has little effect on power. At lower ES, we observe a large loss of power for alleles with less than 0.3 allele frequency. **b** Distribution of mean ± standard deviation of the chloroquine $IC_{50}$ from five biological replicates of 56 progeny and two parents from the NF54 × NHP4026 cross. **c** QTL mapping of mean chloroquine $IC_{50}$ (ES = 0.84) in (**b**) results in a LOD score of 31.13 and a genome-wide $p$ value of 0.0007 demonstrating that despite extreme segregation distortion, QTL can be detected for experimental phenotype data with high ES.

the distorted locus, especially for phenotypes with small ES (Fig. 7). For phenotypes with large ES and for large numbers of progeny, SD has little effect on power; however, as the number of progeny decrease, a substantial loss of power occurs as the degree of SD increases (Fig. 7). For an ES of 0.8, we can detect associations for loci with extremely skewed allele frequency with 30 progeny. For an ES of 0.4, 50 progeny can detect an association for moderately skewed loci, while 84 progeny are necessary to detect an association at extremely skewed loci.

In progeny from the NF54 × NHP4026 cross allele frequencies range from 0.05 to 0.31 and 0.69 to 0.85 (Fig. 3) in regions with significant SD ($p < 0.001$). Consistent with our simulations, we are able to map the chloroquine drug response to the locus containing *pfcrt* ($p = 0.0007$, Fig. 7c) despite the extreme SD at this locus (NHP4026 allele frequency of 0.05). In contrast, in

MKK2835 × NHP1337, allele frequencies of the NHP1337 alleles range from 0.3 to 0.7 (Fig. 3). At these allele frequencies, we see little loss of power indicating that power and mapping resolution are expected to be consistent across the genome.

## Discussion
A well-designed genetic cross can provide high power to dissect complex associations between genotype and phenotype. We demonstrate that targeted crosses between clinical isolates can be rapidly generated and outperform all previous crosses in their size, mapping power, and precision. Furthermore, the uncloned bulk populations from these controlled genetic crosses provide a powerful and complimentary resource for bulk selection analysis/ linkage group selection (BSA/LGS) to identify loci linked to

phenotypes of interest as demonstrated in rodent malaria[38–42] and more recently in *P. falciparum*[24,43]. However, bulk approaches cannot be used to investigate epistatic interactions between loci and collecting -omic data to allow for integrative analyses still requires cloned unique recombinant progeny[44]. We anticipate future linkage analysis with *P. falciparum* will utilise both BSA/LGS and experiments with cloned recombinant progeny to maximzie efficiencies.

Using our progeny arrays and simulated phenotypes, we have shown that traditional QTL mapping with 84 unique recombinant progeny can detect associations for phenotypes with effects sizes (ES) as low as 0.2 and strong epistatic interactions. We can also map phenotypes with small to modest effect sizes more precisely. For instance, at an ES of 0.5 increasing our mapping population from 30 to 84 progeny, we can eliminate 40 candidate genes (Supplemental Fig. 4). For phenotypes with large ES of 0.8, similar to chloroquine resistance, with 84 progeny we can map to a region containing only eight genes. These substantial reductions in number of candidate genes have a large impact on our ability to determine causal mutations, drastically reducing the effort required for validation studies using CRISPR/Cas9-based technology. Furthermore, our ability to generate further genetic crosses between the same two parents of interest is unparalleled, allowing us to potentially isolate 100's of unique recombinant progeny for analysis further improving power and mapping precision.

To maximize recovery of unique recombinant progeny from each cross we initiated multiple cloning rounds per cross. Interestingly, each cloning round produced almost entirely unique sets of progeny indicating that our cloning efforts greatly undersampled the total population of recombinant progeny available (Supplemental Fig. 5). We showed that cloning as early as possible after establishing in vitro culture was important to maximize recovery of unique recombinant progeny. Notably, we minimized the potential for additional loss in diversity during cryopreservation by freezing immediately after mouse exsanguination and cloning within 48 h of thaw. We captured similar proportions of unique recombinant progeny cloning directly ex vivo or from cryopreserved cultures.

We observed different levels of selfing in our allopatric and sympatric crosses. In the 3D7 × HB3 cross selfed progeny were observed at expected ratios in oocysts[45,46] and early in blood stage culture, but at lower than expected ratios among clones when cloning was begun 32 days after the liver stage-to-blood stage transition[47,48]. Similarly, in the 7G8 × GB4 cross, only 29 of more than 200 (14.5%) individual clones were due to selfing[28]. However, our sympatric MKK2835 × NHP1337 cross produced many selfed progeny. Interestingly, NHP1337 dominated the selfed progeny almost entirely, consistent with bulk allele frequencies in samples taken at similar times[24]. While we aimed to infect mosquitos with equal number of MKK2835 and NHP1337 gametocytes, the unequal selfing rates may reflect an imbalance in the initial gametocyte ratio or in gametocyte viability between MKK2835 and NHP1337. It is also possible that there are inherent difference in selfing rates between MKK2835 and NHP1337, although both lines successfully selfed in mosquitoes infected with only one parent (Supplemental Table 1). Bulk segregant analyses of these same populations indicate that these selfed clones were outcompeted over time under in vitro culture conditions;[24] this may also have been the case in the previous 3D7 × HB3 cross[46,47].

In contrast, we observed few selfed progeny in our allopatric NF54 × NHP4026 cross. Both NF54 and NHP4026 readily infected mosquitoes and had higher parasitemias at time of mouse exsanguination when used alone to inoculate mosquitos than when pooled, with NF54 typically giving robust infections

(Supplemental Table 1). Further experiments will be necessary to understand why the NF54 × NHP4026 cross generated so few selfed progeny.

Segregation distortion is common in genetic crosses and often more extreme when more distantly related parents are crossed. For instance, interspecific crosses result in SD more often and with more severe distortion than intraspecific crosses[49,50]. All previous *P. falciparum* genetic crosses were between allopatric parasite lines, and show significant SD (Supplemental Fig. 3). Similar to previous *P. falciparum* crosses, our allopatric cross had regions of significant SD that were consistent across replicates. Conversely, the sympatric MKK2835 × NHP1337 cross was the first *P. falciparum* genetic cross to show relatively even inheritance patterns across the genome with no significant SD. One possible explanation for SD is that natural selection may act against unfit allele combinations causing a deviation from expected mendelian ratios[51]. It is also possible that there are prezygotic barriers, such as barriers to gamete recognition, between distantly related parents or that there is selection on alleles that show adaptation to local vectors during development in the mosquito. To determine where segregations distortion arises, we examined allele frequencies for progeny of the NF54 × NHP4026 cross by cloning rounds initiated after different periods of in vitro culture. The chromosomes 7 and 13 peaks are present in all cloning rounds even when cloning immediately upon mouse exsanguination. The chromosomes 12 and 14 peaks arise later during in vitro culture. A recently completed bulk analysis of an independent replicate of this cross replicates the chromosome 7, 12, and 14 peaks but not the chromosome 13 peak[43]. The bulk analysis adds further resolution on how selection acts over time; the chromosome 7 peak is not present at mouse exsanguination but arises immediately on transition to in vitro culture while the chromosome 12 and 14 peaks arise later during in vitro culture. Therefore, our results indicate that SD is driven by selection at the blood stage in our crosses.

In the NF54 × NHP4026 progeny, the SD peaks contain candidate genes that are related to drug resistance or immune evasion. The SD peak on chromosome 7 includes *pfcrt* which is known to carry a substantial fitness cost on some genetic backgrounds[34]. Although NHP4026 grows well in in vitro culture[52] (even outcompeting NF54 in co-culture experiments[52]) it is clear that inheriting the *pfcrt* NHP4026 allele contributes a large fitness cost that is evident upon transition to in vitro culture.

The SD peak on chromosome 14 is also predominantly inherited from NF54 and contains *pfarps2* which has been associated with artemisinin resistance in GWAS studies and is thought to contribute to a permissive background for development of artemisinin resistance[7]. While NHP4026 is *pfk13* WT it does have a slow clearance phenotype (Table 1). We plan to explore whether *pfarps10* has a fitness cost in this genetic background. Interestingly, while we see no SD in MKK2835 × NHP1337 among the cloned progeny, we do see selection against NHP1337 alleles on chromosome 14 centered on *pfarps10* over time in an uncloned bulk culture of MKK2835 × NHP1337 cross F$_1$ progeny[24].

The SD peak on chromosome 12 is most commonly inherited from NHP4026 and includes *pfmrp2*. An overlapping region containing *pfmrp2* is selected for the MKK2835 allele in an uncloned bulk culture of MKK2835 × NHP1337 cross F$_1$ progeny[24]. *Pfmrp2* has been associated with chloroquine, mefloquine, and piperaquine response in vitro and parasite clearance[35] in Thai isolates and we speculate it may have a fitness cost in vitro. The role *pfmrp2* plays in drug resistance is still unclear and these genetic crosses may help elucidate its function.

The SD peak on chromosome 13 contains both *pf47* and *pfs45/48*, two 6-cys protein-encoding genes that are highly polymorphic

in natural populations and are thought to be under selection because of their roles in gamete recognition and evasion of the mosquito immune response[53,54]. It is thus possible that *pf47* and/or *pfs45/48* play a key role in SD in distantly related parasites with different *pf47* and/or *pfs45/48* alleles. Indeed, we observed no significant SD in the MKK2835 × NHP1337 cross and also observed no selection over time on chromosome 13 in the bulk segregant experiment using the MKK2835 × NHP1337 bulk progeny[24]. Interestingly, in a recently completed bulk analysis of an independent replicate of the NF54 × NHP4026 cross selection on the same region of chromosome 13 does not reach genome-wide significance[43]. Further work will be needed to determine the repeatability of selection on chromosome 13 and the source of selection.

SD loci traditionally have been excluded in genetic mapping studies to avoid loss of power to detect real effects (type II error, false negative) and the potential to detect false positives (type I error)[55]. Excluding distorted loci from analysis would be particularly problematic in *P. falciparum* because all previous crosses had regions of significant SD. Using SD loci in mapping studies is possible, although it reduces power to detect QTLs. Through simulations, we have shown that SD drastically decreases power to detect QTL for phenotypes with medium and small effect size. For phenotypes with large effect size even extreme SD does not decrease power substantially. In NF54 × NHP4026 progeny we are able to correctly map the chloroquine drug response to the locus containing *pfcrt* despite an NHP4026 allele frequency of less than 0.05 at *pfcrt* and only three progeny showing a chloroquine-resistant phenotype. For phenotypes with small to moderate effect sizes, care is required in interpreting negative QTL results, especially when mapping in small progeny sets. When QTL and SD loci coincide, false negatives will lead us to miss true associations between phenotypes and genetic variants. This problem with power will be amplified when attempting to map -omics phenotypes where multiple testing corrections must be employed.

In conclusion, we believe that the use of the human liver-chimeric FRG NOD huHep/huRBC mouse to generate genetic crosses will revolutionize *P. falciparum* quantitative genetics. It is now feasible to generate crosses on demand to study the genetic architecture of emerging phenotypes. We can also use complex cross designs to further improve power to detect associations for phenotypes where a genetic variant only controls a small amount of variation.

## Methods

The genetic cross pipeline is outlined in Fig. 1 and our previous methods publication[23]. The pipeline begins with the identification of isolates with interesting phenotypes that have been culture adapted and cloned. Cloned isolates are then evaluated for their ability to produce gametocytes. Cloned isolates were maintained under standard culture conditions in human erythrocytes (O$^+$ at 5% hematocrit) suspended in complete media (RPMI-1640 with 2 mM L-glutamine, 25 mM HEPES, 50 μM hypoxanthine and 10% A+ human serum) at 37 °C under an atmosphere of 5% $CO_2$, 5% $O_2$ and 90 $N_2$. Gametocyte cultures were initiated at 0.8–1% parasitemia. Cages of 200–250 adult female *Anopheles stephensi* mosquitos were infected using gametocytes of either parental strain or both parental strains as described in Supplemental Table 1. For each cage 10 mosquitos were sacrificed on day 10 to measure infection prevalence and number of oocysts/mosquito. Cages with the best infection rates were then used to infect the FRG NOD huHep mice via IV injection or directly by mosquito bite (MB) (Supplemental Table 1). The FRG NOD huHep mice were injected with 400 μL of packed O$^+$ human RBCs on days 6 and 7 after infection. The mice were exsanguinated 4 h after the final huRBC injection to recover *P. falciparum* infected huRBCs. The recovered blood was washed 3 times with 10 mL of complete media, pelleted at 200 g and resuspended in an equal volume of O$^+$ huRBCs and complete media at 2% hematocrit. Thin smears were made to estimate parasitemia and cloning via limiting dilution was initiated as soon as possible (cloning procedure in Supplemental File 1). Bulk culture was also cryopreserved for future cloning.

We made several adjustments to maximize recovery of progeny from the genetic crosses, including completing independent replicates of the crosses and cloning via limiting dilution directly from the transitioned blood removed from the FRG NOD

huHep mouse. In addition, the transition to in vitro culture was carried out using media containing Albumax rather than human serum. We observed successful expansion of the transitioned cultures in both serum-containing and Albumax-containing media, but downstream limiting dilution cloning failed to yield the expected number of clones if carried out using serum. We, therefore, cloned and expanded the transitioned blood stage culture in media containing Albumax. Screening for clones was carried out using the Phusion Blood Direct PCR Kit (Thermo Scientific). Specific methodological information for each replicate of each cross is provided in Supplemental Table 2.

The study was performed in strict accordance with the recommendations in the Guide for the Care and Use of Laboratory Animals of the National Institutes of Health (NIH), USA. All of the work carried out in this study was specifically reviewed and approved by the Seattle Children's Research Institute IACUC under ACU00598 Harnessing the power of genetic crosses.

**Identifying positive clones**. Beginning at week 2 post cloning and continuing on weeks 3, 4, and 6, the Phusion Blood Direct PCR Kit (Thermo Scientific) was utilized to identify positive clones. This kit is very sensitive, detecting positive parasitemia using only 1 μL of infected culture streamlining our detection of positive clones. We are able to detect parasites at parasteimias as low as 0.001%[56]. Wells with a CT score of 25 or lower were moved into 24 well plates and assigned a unique alphanumeric identifier. These cultures were then expanded to 1 mL for further expansions and cryopreservation of stocks and material for genotyping using the assigned unique identifier. A protocol for this screening method is available in the S1 File.

**Genotyping**. All progeny of NF54 × NHP4026 were initially genotyped via microsatellite markers to identify unique recombinants. The progeny isolated in cloning rounds 1 and 2 or replicate 1 of the NF54 × NHP4026 were genotyped at 17 MS markers. The progeny isolated in cloning round 3 of NF54 × NHP4026 were genotyped at 8 MS markers. Primers for each MS marker used are listed in Supplemental Data 4. For cloning rounds 1 and 2, full genome sequencing was performed for all unique recombinants. For cloning round 3 and all other crosses all potential recombinant progeny were fully sequenced.

**Preparation and sequencing of progeny**. DNA was extracted from 35 to 50 μL of packed red blood cells using Quick DNA Kit (Zymo). Libraries were prepared with ¼ reaction volumes of the KAPA HyperPlus DNA Library Kit and 20–50 ng of extracted DNA according to manufacturer directions with slight modifications. Fragmentation time was 26 min; adapter ligation was increased to 1 h; PCR was performed for 7 cycles; and size selection was performed post PCR using full volume methods. We used KAPA Dual-Indexed Adapter Kit, adding 7.5 μM adapter to the appropriate well. Samples were measured for DNA quantity using the QBit BR DNA Kit. Samples were then pooled for sequencing based on their QBit measurements to normalize input. The pooled sample was quantified using the KAPA Library Quantification Kit, and adjusted to 2–4 nM with 10 mM Tris-HCl, pH 7.5–8.0 (Qiagen) for sequencing on Illumina platforms. The pool was also run on the Agilent Tape Station using the D1000 BR Kit to assess sample size and lack of primer dimers. Pools were run on the Illumina HiSeq 2500 or Illumina NextSeq for 2×100 bp run.

We aligned raw sequencing reads to v3 of the 3D7 genome reference (http://www.plasmodb.org) using BWA MEM v0.7.5a[57]. After removing PCR duplicates and reads mapping to the ends of chromosomes (Picard v1.56) we recalibrated base quality scores, realigned around indels, and called genotypes using GATK v3.5[58] in the GenotypeGVCFs mode using QualByDepth, FisherStrand, StrandOddsRatio VariantType, GC Content and max_alterate_alleles set to 6. We recalibrated quality scores and calculated VQSLOD scores using SNP calls conforming to Mendelian inheritance in previous genetic crosses, and excluding sites in highly error-prone genomic regions (calls outside of the "core genome"[21]).

**Filtering high quality SNP variants**. The.vcf file containing parents, potential progeny, and all high quality SNPs were processed in R using the vcfR library. Initially SNP filters were based on the parental distributions; only homozygous, bi-allelic parental SNPs with high coverage (≥10) and high quality scores (GQ ≥99) were retained. Next, low quality SNPs across parents and progeny were filtered with a VQSLOD < 2.5. This final SNP set was defined as our high quality SNP set for further analysis.

**Filtering progeny**. In *P. falciparum* crosses to produce the F$_1$ mapping population, it is necessary to filter out potential progeny that are non-clonal and repeated sampling of the same genotype. Initially, potential progeny with more than 80% missing data were removed from further analysis.

**Identifying and filtering non-clonal progeny**. Since *P. falciparum* parasites are haploid throughout the entirety of the human portion of their life-cycle including the intraerythrocytic stage during which they are cloned we expect that clonal infections should have predominantly homozygous SNP calls except for rare instances of sequencing error. In contrast, non-clonal infections where the mixture

contains full siblings or full siblings and parent genotypes would have contiguous regions with high numbers of heterozygous SNP calls at above the rate expected from sequencing error along.

The sequencing error rate was estimated for each cross as the mean from a distribution of percent heterozygous SNP calls across all potential progeny (Supplemental Fig. 6). Assuming true sequencing errors follow a Poisson process with $\lambda$ = % sequencing error, then the expected distance between sequencing events as $1/\lambda$. To identify non-clonal samples we counted heterozygous SNP calls across the genome in a sliding window of size $1/\lambda$ and using a Poisson Distribution with $\lambda$ = % sequencing error calculated the probability of getting at least the observed number of heterozygous SNP calls in each window. These probabilities were adjusted for multiple testing based on the number of windows in the genome and the adjusted probabilities were plotted as a heatmap (Supplemental Fig. 6). Samples with windows with adjusted probabilities <0.05 were designated as non-clonal and filtered from the final progeny set.

**Phasing of clonal progeny**. A matrix of phased genotypes was constructed for parents and clonal progeny for all high quality SNPs. In each cross the drug sensitive parents (NF54 and MKK2835) were coded as 0 while the drug resistant parent (NHP4026 and NHP1337) were coded as 1. Progeny SNPs that matched the drug sensitive parent's SNPs were coded as 0 while SNPs that matched the drug resistant parent's SNPs were coded as 1. Heterozygous SNPs were coded as missing.

**Identifying unique recombinants**. Our high quality phased dataset for clonal progeny was formatted for the qtl package in R and loaded as a genetic map. Genotype similarity scores were computed using the comparegeno function. A similarity score cut-off of 0.9 was used to define clusters of genetically distinct recombinant progeny. Individual progeny were selected from each cluster of genetically similar progeny using igraph in R. Only unique recombinant progeny and parents were retained to create a final dataset of all SNPs. Unique recombinant progeny cryopreserved stocks were thawed and expanded within two weeks to created generation 1 stocks for later phenotyping and long term storage.

**Visual recombination map construction**. 5 kb windows were defined across the core genome to construct a heatmap depicting a recombination map for each cross. For each progeny, in each 5 kb window the most common parental genotype was determined, if a window contained only missing data then it was filled if the next window with data had a matching genotype to the previous window with data, otherwise it was left missing.

**Defining informative markers**. All phased genotype data for clonal, unique recombinant progeny and parents were loaded into R qtl as a genetic map. The findDupMarkers function was used to identify clusters of markers with identical genotype data and the most central marker with the least missing data from each cluster was retained in a set of informative markers. Sets of markers were excluded if they spanned less than 100 bp. Non-duplicated markers were included in the set of informative markers if they had a stronger correlation to flanking marker sets than the flanking marker sets did to each other. This set of informative genotype markers for all clonal, unique recombinant progeny was used for all subsequent analysis and figures. The entire filtering pipeline is available on github (https://github.com/kbuttons/P01_ProgenyCharacterization) with documentation.

**Genetic map construction**. For each cross the set of informative genotype markers for clonal unique recombinant progeny was coded as "a" for the sensitive parent and "b" for the resistant parent and "–" for missing data and loaded into Join-Mapv4.1. Population type was set to HAP1 and the Kosambi mapping function was employed in generating each genetic map. All other parameters were initially set to defaults, however, to account for the systemic SD observed in the NF54xNHP4026 cross it was necessary to expand the population threshold ranges such that the independence LOD ranged from 1.0 to 15.0, the independence $P$-value from 1.0e$-3$ to 1.0e$-5$, the recombination frequency from 0.250 to 0.001 and the linkage LOD from 3.0 to 15.0. This change in parameters allowed us to differentiate between SNP markers with similar distortion patterns that were known to be physically located on different chromosomes.

**Drug response profiles**. Parasites were cultured under standard culture conditions in human red blood cells suspended in complete medium (CM) containing RPMI 1640 with L-glutamine (Invitrogen Corp.), 50 mg/L hypoxanthine (Sigma-Aldrich), 25 mM HEPES (Cal Biochem), 0.5% Albumax II (Invitrogen Corp.), 10 mg/L gentamicin (Invitrogen Corp.) and 0.225% NaHCO$_3$ (Biosource) at 5% hematocrit. Cultures were grown separately in sealed flasks containing 5% CO$_2$, 5% O$_2$, and 90% N$_2$ at 37 °C. Drug tests were performed when cultures were at least 80% ring with at least 96 h between tests. IC$_{50}$ curves were generated for parent lines for chroloquine, piperaquine, and dihydroartemisinin and for chloroquine for 56 NF54 × NHP4026 progeny by standard [$^3$H]-hypoxanthine incorporation assays[3]. eRRSA, an in vitro artemisinin resistance surrogate, were measured following our published methodology[56].

**Power analysis**. Progeny from NF54 × NHP4026 were used to estimate power under three different scenarios, one genetic locus contributing to phenotypic variation, 2 loci with additive contributions to phenotypic variation and 2 loci with epistatic interaction controlling phenotypic variation. All models were simulated for the full F$_1$ progeny set with $N = 84$ and for subsamples with $N = 30, 40, 50, 60,$ and 70. Under the one locus model, a phenotype was simulated as either a single replicate value or the average of 5 replicates at effect sizes ranging from 0.1 to 0.8. Under the two additive loci model, a phenotype was simulated as either a single replicate value or the average of 5 replicates for effect sizes for each locus ranging from 0.1 to 0.4. Under the two epistatic loci model, the first locus controlled whether a trait was present in an on/off fashion and the second locus controlled the level of the phenotype (i.e., locus 1 is necessary to be drug resistant and locus 2 controls the level of resistance) and the main effects of both loci ranged between 0.1 and 0.4. A set of markers with 1:1 mendelian inheritance patterns were used as the 1 or 2 loci in the models. All QTL mapping was performed with r qtl. For each simulation, significance thresholds were defined based on 1000 permutations. True positives were defined as a LOD peak that meant the $\alpha = 0.05$ significance threshold and whose 1.5 LOD interval contained the actual marker used in the model.

**Segregation distortion power analysis**. This analysis was similar to the 1 locus model in the previous section. In these simulations effect sizes were calculated based on balanced inheritance and levels included 0.2, 0.3, 0.4, 0.6, and 0.8. All markers were categorized by their allele frequency and sorted into bins for each level of allele frequency skew (i.e., 0.89 to 0.91 and 0.09 to 0.11 were in the 0.1 bin which represented the most skewed alleles in this analysis). QTL mapping, significance levels and definition of true positives were the same as in the power analysis above.

**Statistics and reproducibility**. Drug response profile data (IC$_{50}$ and eRRSA) are reported as the mean and standard deviation of 5 individual biological replicates (drug tests were loaded from cultures grown for at least 2 cycles independently).

**Ethics approval and consent to participate**. The study was performed in strict accordance with the recommendations in the Guide for the Care and Use of Laboratory Animals of the National Institutes of Health (NIH), USA. To this end, the Seattle Children's Research Institute (SCRI) has an Assurance from the Public Health Service (PHS) through the Office of Laboratory Animal Welfare (OLAW) for work approved by its Institutional Animal Care and Use Committee (IACUC). All of the work carried out in this study was specifically reviewed and approved by the SCRI IACUC under ACU00598 Harnessing the power of genetic crosses.

**Availability of biological material**. All unique recombinant progeny generated in this study can be provided upon request subject to a reasonable fee to cover shipping, reagents, and cost of creating progeny stocks. We also encourage collaborative projects using this research.

**Reporting summary**. Further information on research design is available in the Nature Research Reporting Summary linked to this article.

## Data availability

All raw sequencing data have been submitted to the NCBI Sequence Read Archive (SRA, https://www.ncbi.nlm.nih.gov/sra) under the project number of PRJNA524855 as SRR11835641 to SRR11850234. VCF files and all data files used to produce figures are available at https://github.com/kbuttons/P01_ProgenyCharacterization.

## Code availability

All analysis codes are available at Zenodo (DOI:10.5281/zenodo.4591063) and github https://github.com/kbuttons/P01_ProgenyCharacterization.

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

## Acknowledgements

This work was funded by National Institutes for Health (https://www.nih.gov) grant P01 AI127338 (to M.F.), NIH grant R21 AI133369 (to A.V.), NIH grant R01 AI110941-01A1 (to I.H.C.), and NIH grant R37 AI048071 (to T.J.C.A.). The funders had no role in study design, data collection and analysis, decision to publish, or preparation of the manuscript. We would like to thank Jasmine Clark for help with progeny cloning. We would like to acknowledge members of the Ferdig lab, Anderson lab, Cheeseman lab, Vaughan lab, Kappe labs, and Emrich lab for helpful discussions.

## Author contributions

K.B.S., S.K., X.L., T.A., M.F., S.E., A.V., and I.C. contributed to the conceptualization of the study. K.B.S., S.K., N.C., M.H., C.J., L.C., R.S., and I.C. curated data. K.B.S., X.L., J.R. S., S.E., and I.C. performed the formal analysis of the data. S.C.K., T.A., M.F., S.E., A.V., and I.C. acquired funding for this study. S.K., N.C., M.H., C.J., L.C., S.Y.K., R.S., D.S., M. M.W., and A.V. contributed to improved methodology for generation of a recombinant progeny population, cloning of individual progeny, and subsequent genotyping. K.B.S., S. K., N.C., L.C., M.M.W., A.V., and I.C. contributed to project administration. F.N., S.H.K., T.A., J.R.S., M.F., S..E, A.V., and I.C. contributed resources to this study. K.B.S., S.E., and I.C. developed all analysis scripts used in this study. K.B.S., S.K., N.C., C.J., L.C., S.Y.K., R. S., D.S., S.H.K., T.A., J.R.S., M.F., S.E., A.V., and I.C. contributed to the validation of all new methods employed in this study. K.B.S. produced all visualizations. K.B.S., J.R.S., S. E., A.V., and I.C. wrote this article. All authors reviewed and edited this article and approve the final submitted version.

## Competing interests

The authors declare no competing interests.
