## [Peer Review File · Communications Biology]

Reviewers' comments:

Reviewer #1 (Remarks to the Author):

The authors of the manuscript: "Surprising variation in two malaria genetic crosses using humanized mice: implications for genetic mapping" describe the generation and outcome of two genetic *Plasmodium falciparum* crosses in the humanized mouse model. While the authors have previously reported on both crosses, they provide a more detailed analysis of the statistical power to determine underlying genetic variation and simulate the number of progenies needed to detect linkage for traits with different effective sizes, even detecting epistatic interactions. Generating this massive amount of new progeny from 2 different crosses is a tour de force and more than doubles the number of parasite lines generated in previous crosses using primates. This is generating a valuable resource for the malaria community. The humanized mouse model allows the much faster generation of new crosses without the need of primates. The authors were able to reproduce the observed segregation distortion of the NF54x NHP4026 cross and generate progeny of a new cross between two highly related recent parasite isolates. The authors provide detailed observation about the segregation of the progeny and find high selfing in one cross and segregation distortion in the other cross. The high number of unique progenies generated here however, allows the authors to still detect linkage of various effective sizes.

Overall, the data is well described and presented but a few additions would help the reader to understand the data better. The initial strains are not described well and it should be stated more clearly and earlier in the manuscript that part of the data on the Thai cross has been published previously. There are discrepancies between the tables and the text and the figure legends are not always providing enough information to understand the data they are showing. The authors refer to EC50 data that is not shown in this manuscript.

Disclaimer: I do not have the right background to comment on the accuracy or appropriate generation of the simulation data.

In the introduction, the authors claim that this is the first report of such data but they have published part of their study already (PMID: 31609965).

Line 165: The authors give the reference for the NF54-GFP_{Luc} x NHP4026 cross (line 158) but not for the previously described MKK2835 x NHP1337 cross. Was this done on different occasions or is it from the same experimental setup? This needs to be clarified.

I have a hard time reconciling the Supplementary Table 1 and the description in the text. There are 6 replicates for NF54xNHP4026 in the table but only 3 have been used for infections. Please also indicate in the table which cage has been used. There is an auto-population error for MKK2835xNHP1337. Again, there are 8 cages in the table but only 4 are referenced in the manuscript and only one has been used for the cross.

How are the prevalence and oocyst numbers determined if the sporozoites are used for infection?

Line 183: please define RFLP and MS markers

Line 194: How many were selfed vs non-clonal?

Line 201: What is a cloning round? Different mouse? Different whole infection cycle? Different plate? This does not become clear until the next section: Inbreeding, Outbreeding and Plastid inheritance. And it is not clearly defined. Maybe including it in the schema in Figure 1 would be helpful. Please provide a clear definition before using the term.

Line 230: In the text it says that there were 3 selfed NF54 and no NHP4026 but there are only 2 dots in figure 2A for NF54 and one for NHP4026 in 2B. Maybe I don't understand what data has been included in Figure 2.

Line 238: How were the parasites kept for the additional days to 14 and 19 days? Was there a

bottle neck (eg most of the parasites taken out for the first round of cloning)? Or was only a small amount removed for cloning and the bulk kept? Was the culture split or was the parasitemia low enough?

Line 298: There is not A and B in figure 4.

A better description of the strains used here would be useful. Maybe a table with the drug resistance pheno- and genotypes?

Line 313: 3 offspring are CQ resistant which ones? Where are the EC50 values? What is their pfCRT status?

Line 325: It has been shown by Carolina Barillas-Mury group that pf47 is important for transmission in the mosquito stages due to evasion of the mosquito immune response (PMCID: PMC5519742, 29229188, PMC5167319). What mosquito strain has been used for this cross?

Line 334: Just to clarify, the bulk analysis published previously and these clonal lines are not from the same cross?

Line 337: This is a very vague statement. There is overlap between all of the crosses ever done or just the bulk? Also, it might be helpful to show it in the figure similar to figure 4. In the Sup figure 2 legend it says that there is no significant overlap. Include the references for the crosses. Is A bulk or clones? Please clarify what you are comparing in the text.

How is the effective size of a phenotype established? What are some examples from the literature?

Discussion:

Lines 466/7: The authors claim that this is the first report of the MKK2835 and NHP1337 cross, yet they published on before. I guess it's the first time they looked at the individual progeny of that cross.

Line 471: The high selfing proportion of NHP1337 was observed in the previous paper but was lost in vitro over time.

I am surprised that the sampling over time does not include more of the same genotypes. The authors explain this by under-sampling the possible recombinant progeny. In lines 163/4 it was speculated how many possible new genotypes could have been used to infect the mice used here. Could you use some statistics to back up this claim? You sample about 10% of the possible new progeny. What would be the expected overlap?

Was there a loss over time for k13 mutants like in the previous bulk population?

Line 532: typo be not by

Line 535: Please indicate what the expected ratio is.

Line 538: The authors speculate that the selfed progeny will be outcompeted in vitro growth over time. This would agree with their previously published data on the bulk analysis. However, here it looks like MKK2835 is picked up in round 1 and 2 with more clones in round 2 (Fig. S1 B).

Although the numbers don't seem to agree with sup table 2 and too many unique recombinants are reported for cloning straight from mouse. It's 18 in the table and a total of 78 but from the figure it looks more like 70 and the total is well above 100.

It would be interesting to look specifically at the k13 locus on all clones to see if one variant increases over time. As the cloning was done shortly after exsanguination it might not have been enough time in vitro for selection. Have the two parental lines ever been co-cultured in vitro to detect different growth advantages?

When discussing the segregation distortion between the different lines and the genes potentially responsible for this difference it would be useful to actually know the SNPs in these genes and if

the isolates are different from each other at this specific locus. The fitness cost for each segment is primarily a function of viability in the mosquito and liver stages as there is only a short amount of in vitro growth and no competition with other genotypes due to the cloning. What was the time spread for clones to come up? This could be a useful phenotype to look for genes important in in vitro growth.

I'm wondering how much segregation distortion is a function of in vitro growth for the highly related parasite lines as well. For the bulk experiment, there was no distortion at the early blood stage but began to manifest after about 2 weeks in culture.

Figures and Tables:

Figure 1. Please don't use a picture of an Aedes mosquito. They do not transmit malaria.

Figure 2: I was confused by the different colored cell-like shapes around the parasite dots. Instead of making it clear that they contain clonal parasites they reminded me of hepatocytes. Maybe just using circle around it or joining them by a line? Or maybe have the circle size indicate the number of progenies detected? Also, "*" denotes the only observed repeat sampling event between cloning rounds" Shouldn't it be next to the two dark blue and one light blue dot? Using yellow for the parents is also confusing as it's not clear in which round it happened. Putting the name next to it will be enough. There is also a dark blue spot in the light blue cluster for MKK2835 cluster. It's hard to see the difference between round 1 and round 2 as the two blues are very similar. I like the summary of 2C. Maybe these colors could be used in 2A and B to indicate the different occasions? E.g. yellow circles around repeat genotypes within rounds, orange around selfed progeny etc.

SF1: As the colors for A and B are the same they don't need to be repeated twice in the legend.

Figure 4: I do not see any black dots in the upper figure even though there is a legend for them. From which cloning round is this data? Is it all combined? Is there a difference between cloning rounds?

Figure 7B mentions the EC50 values of 35 clones to CQ used for QTL analysis. Please provide the values in a table. Also what is the ES of CQ resistance?

Sup table 2: How were the numbers for unique recombinant progeny determined for the Thai cross? I don't see how the numbers add up. The days before cloning is confusing for the Thai cross. Why is the crypreserved source considered 2 days before cloning but round 2? How long was it in culture before sub-cloning?

Supplementary cloning protocol. At what day after starting the cloning plate is the PCR done? What is the minimum parasitemia detected? Was there ever a dilution series done to determine sensitivity? I wonder if you only pick-up fast-growing parasites and throw away all with a growth defect that need longer to come up. Also, the empty wells seem to have relatively high CT values as well.

Reviewer #2 (Remarks to the Author):

Button-Simons et al. describe generation of two *P. falciparum* genetic crosses in the human liver-chimeric FRG NOD huHep mice. Four *P. falciparum* genetic crosses in chimpanzees have been published, and the authors have also reported a *P. falciparum* genetic cross using the humanized mice previously; therefore, the procedures for generating progeny from *P. falciparum* crosses are not new. The contributions of this report mainly include: 1, generation of relatively large numbers of progeny from two crosses (60 and 84 progeny, respectively); 2, high resolution genetic maps with observations of inheritance bias (or not); 3, estimates of mapping power, which is important for future genetic mapping studies. The manuscript is generally well written, expect discussions are long repeating many results. This study represents an important contribution to malaria

genetic mapping field, including generation of large numbers of progeny.

Some minor points;

--Abstract: ">140 progeny", better 144 progeny.

--Line 82-82: Spell out the genes.

--Line 184: the refs are not accurate. The correct refs should be Wellems et al., Nature 1990 345; Su et al., Cell 1997, 91. These are the papers initially used RFLP and microsatellites initially.

--Line 221-225. Can just say the same as A for MKK2835 x NHP1337 cross.

--Line 265, "core region": please define the core regions.

--Line 300: Remove (A) because there is no (B). Or add (B) panel.

--Fig. S2. Please provide the source data for the plots of published *P. falciparum* crosses.

--Line 248, "effect size": Is penetrance better? Or functional impact?

--Line 388, "association" and more: For mapping using family data or progeny, linkage is better. Association is more reserved for population based studies.

--Fig. 6, the color lines at the top right corner appear not matching those in the figure.

--Line 429, SD: please spell out.

--Line 608, "by": be.

--Line 773, Physical recombination map: Suggest removing "physical" to avoid confusion with a physical map.

--Line 829: Please provide the approved animal protocol number for the work.

-- References, some with full titles, some are abbreviated.

--The Excels need a title for each table and some explanations. For example, "Days before cloning" in 117502, are the days in culture? Another, 117501, "Post feed exflagellation", I assume that the numbers are percentages?

--The cloning document appears to be an internal protocol; may need some edits for publication.

Reviewer #3 (Remarks to the Author):

The manuscript "Surprising variation in two malaria genetic crosses using humanized mice; implications for genetic mapping" by Button-Simons and colleagues is a generally well-written account of the analysis of two genetic crosses between cloned isolates of *Plasmodium falciparum* performed in humanised mice. The major findings of this work involve a power analysis that was performed in order to determine the number of repeats of a particular cross, and the numbers of cloned recombinant progeny that need to be analysed from them in order to detect the genetic drivers of single or multi-genic phenotypes. Additionally, the authors demonstrate that one of their crosses, that between NF54 (or a clone of NF54, presumably) and a culture adapted field isolate from Asia (NHP4026), produced progeny that exhibited significant variation across the genome with respect to the proportions of alleles inherited from one or other of the parents, whilst another of their crosses, this time between two Asian isolates, had a more uniform proportion of parental alleles across the genome. They also found that in both crosses, progeny inherited the maternally-inherited plastid and mitochondrial genomes from both parents, in contrast to previous crosses in *P. falciparum* in which the progeny appeared to have inherited these genomes from only one of the parents presumably due to unequal representation of parental gametocytes during the production of the cross.

Overall, this manuscript is interesting, and represents the results of a huge amount of work, on which the authors are to be commended. I enjoyed reading it, and learnt from it.

My major criticism relates to the fact that nowhere in the manuscript is mention made of the previous work performed with rodent malaria parasites which covers very much the same ground; how to deal with maximising the investigative powers of genetic crosses in malaria parasites whilst dealing with uneven representations of parental alleles in progeny which are independent of the phenotype under investigation. I refer to Richard Carter and colleagues "Linkage Group Selection" work, which culminated in Abkhallo et al's PLoS Pathogens paper from 2017 in which the genetic drivers of two independent phenotypes, one of which was multigenic, were investigated using multiple crosses between two parental strains. I completely understand that this was in rodent malaria parasites, but the parallels are obvious, and it seems an odd omission from the current

MS.

Indeed, linked to this, throughout the manuscript it was not obvious to me why progeny should be cloned at all. Given that the selection events here should be observable through quantitative-seq LGS (a type of bulk segregation analysis), the extensive time and resources required to clone and genotype individual progeny should be clearly justified. The result given in Figure 4 are the result of producing a cross, cloning large numbers of progeny (months of work), then pooling the individual results from each clone into one analysis. Surely, this could have been done without the need for cloning, and the results would have better resolution for it? This would render the remainder of the MS which deals with power calculations in which the numbers of progeny are a crucial parameter somewhat dispensable.

The above criticism is not critical, and can be addressed by inclusion of a discussion of the advantages of cloning over bulk segregation analysis, and specifically q-Seq LGS, in the discussion and possibly the introduction.

Specific comments:

The Title. Is the variation between crosses really surprising? As the authors point out in the MS, all previous Pf crosses have displayed "segregation distortion". This is presuming that it is the differences in "segregation distortion" that the authors are referring to as being surprising (!). I think a more suitable title should be used, perhaps one which is more descriptive? Something along the lines of "Genetic mapping of Plasmodium falciparum crosses generated in humanised mice", but perhaps less boring.

Results, Lines 163 through 170. The figures quoted here for the maximum number of recombinants from each cross confused me at first, until I realised that they were calculated based on perfect outcrossing, with no selfing. This is highly unrealistic, as in a cross where equal numbers of gametocytes of each parental are provided for the cross, then only 50% of the oocysts would produce recombinants. The only way I can think of to avoid selfing at all, is to use parental clones in which one of the gametocyte sexes is removed from each parent (technically possible, and perhaps an interesting future approach...).

Figure 1 – just a comment but 2/3rds of the time line is taken up with cloning and characterising clones – this could be dramatically reduced if analysis is performed on the uncloned recombinant progeny.

Line 200; It is not obvious here that the "multiple cloning rounds" referred to were sequential, and what time interval separated them. Please include a clearer description here.

Figure 2. This is a nice figure, but giving the clusters different colours, especially colours similar to the "cloning rounds" colours is a bit confusing. Could a black dotted line be drawn around "clusters" instead? Again, the timings of the cloning rounds is crucial information to interpret these results and should be made explicit.

Line 230: Information on the relative proportions of gametocytes of each of the parental strains used in the production of the crosses should be given, as this has a major bearing on the interpretation of the subsequent data. It is crucial that this information is added to the results, either in a table, or mentioned explicitly in the text.

Line 244. The fact that in a perfect cross, 50% of progeny will inherit apicoplast and mitochondrial genomes from Parent A, and 50% from Parent B has never been disputed. The authors set up a bit of a straw man argument here; the fact that two previous *P. falciparum* crosses deviated from this expectation (presumably due to the gametocyte ratios used in their production) was always seen as an experimental anomaly, and not evidence that this is the normal state of affairs. I think it's the sentence "We show here that this is not the case" that is the problem; it gives the impression that people believe that only one parent can contribute these genomes to the recombinant progeny population.

Figure 3. This is a great figure, and almost by itself justifies all the cloning! However, is it possible to split Panel A into two separate panels showing the results from each of the two rounds of crossing for this cross? This would allow the reader to gauge the extent of the similarity between them. The legend of this figure mentions the differences between the parents with respect to their sensitivities to anti-malaria drugs for the first time in the MS; this should be mentioned earlier in the text. There is no mention in the legend of what the black horizontal lines on allele frequency graphs are meant to represent. If these are cut-off points for "significance" when considering deviation from mendelian expectations, then this should be stated along with an explanation of how they were calculated.

Figure 4: I couldn't find any panel labels on the figure embedded in the MS. I'm presuming the whole genome marker graph is panel A. Do the authors have bulk segregation analysis data for this cross at the same points? If so, it would be really interesting to see that data here in comparison to the results presented. Furthermore, as clones were generated from different cloning rounds, sequentially through time, there is the opportunity to show how allele frequencies change through time here; this would illustrate if the selection valleys observed are the result of selection during the blood stage growth of the parasites, or not. Valleys would be expected to get deeper (and peaks higher) through time, if selection was occurring during culture... It would be fantastic to see the data separated like that here. Any peaks and troughs that do not change their heights/depths through time must have been the result of selection pre-in vitro culture, and therefore be the result of mating incompatibilities or selection during the sporogonic cycle... Again, this figure would be much more informative if the recombinant progeny had not been cloned...

The coloured panels at the bottom of the figure are not very useful. Instead of Pf_XXXX numbers, is it possible to give the actual names of the genes (where known)? Otherwise the panels are not really informative. Also, on the version of the figure I saw, they are way too small to read properly.

Could the genes under apparent selection be put in a table in the main text?

Line 320; What does "single genetic variants" mean here?

Figure 5: the power analysis is interesting, but again, is it possible to include a model that does not involve cloning?

Line 382. There are many phenotypes that are controlled by single genes in malaria parasites ("genetic traits" is not a good way to describe "phenotype" in this sentence). Could the authors provide references for multi-gene controlled phenotypes in malaria parasites?

Line 400; Again, these results should be discussed in comparison to LGS, in which multiple genotypes may be analysed in a single (albeit repeated) cross, even if the trait is multigenic.

Lines 438. "Segregation Distortion" I feel the authors may not be considering that the stages of the malaria parasite immediately following fertilisation and up to the appearance of blood stage parasites constitute a long window of potential selection. How do we know that the "segregation distortion" observed in blood stage parasites is not simply the result of selection at the ookinete, oocyst, sporozoite and/or liver stages? Each of these points constitutes a potential bottleneck at which particular genotypes might be favoured over others. This should be considered, and discussed in the MS. Again, it may be possible to separate selection at the mosquito stages from selection at the blood stages by looking for allele frequency movement through time during blood stage culture.

Line 574: typo – ratio instead of 'ratios'

Line 575: Malaria parasites show adaptation to local vectors. The parasites of S. America are easily transmitted by S. American mosquitoes, and not so well by African/Asian mosquitoes. This could account for differences in allele frequencies observed in crosses between isolates from different continents.

Line 596-597. I reread this sentence several times, but could not understand it. Which is the "derived" allele? Could this section be re-written for clarity?

Line 618: Again, the possibility of selection at the sporogonic and liver-stages should be considered.

Line 662 Perhaps the word "that" has been accidentally inserted?

We thank the reviewers for their comments, these were invaluable in adding greater clarity and accuracy to the manuscript. We have provided a point-by-point response to the comments raised by the reviewers in the blue text below.

Referee expertise:

Referee #1: *P. falciparum* genomics

Referee #2: Host-malaria parasite interaction, and genetic cross generation

Referee #3: Mixed-malaria parasite co-infections

Reviewers' comments:

Reviewer #1 (Remarks to the Author):

The authors of the manuscript: "Surprising variation in two malaria genetic crosses using humanized mice: implications for genetic mapping" describe the generation and outcome of two genetic *Plasmodium falciparum* crosses in the humanized mouse model. While the authors have previously reported on both crosses, they provide a more detailed analysis of the statistical power to determine underlying genetic variation and simulate the number of progenies needed to detect linkage for traits with different effective sizes, even detecting epistatic interactions. Generating this massive amount of new progeny from 2 different crosses is a tour de force and more than doubles the number of parasite lines generated in previous crosses using primates. This is generating a valuable resource for the malaria community. The humanized mouse model allows the much faster generation of new crosses without the need of primates. The authors were able to reproduce the observed segregation distortion of the NF54x NHP4026 cross and generate progeny of a new cross between two highly related recent parasite isolates. The authors provide detailed observation about the segregation of the progeny and find high selfing in one cross and segregation distortion in the other cross. The high number of unique progenies generated here however, allows the authors to still detect linkage of various effective sizes.

Overall, the data is well described and presented but a few additions would help the reader to understand the data better. The initial strains are not described well and it should be stated more clearly and earlier in the manuscript that part of the data on the Thai cross has been published previously. There are discrepancies between the tables and the text and the figure legends are not always providing enough information to understand the data they are showing. The authors refer to EC50 data that is not shown in this manuscript.

Disclaimer: I do not have the right background to comment on the accuracy or

appropriate generation of the simulation data.

1. In the introduction, the authors claim that this is the first report of such data but they have published part of their study already (PMID: 31609965).

Although we have reported on these crosses before, the submitted manuscript is the first complete description of the progeny of this cross. In the quoted publication we briefly refer to a part of the data described here to support the likelihood of a high degree of selfing in progeny pools used for bulk segregant analysis. This description was restricted to summary data, and no progeny level information was reported. We have added references to this previous publication on line 168, 177, 368 and 588.

2. Line 165: The authors give the reference for the NF54-GFP^{Luc} x NHP4026 cross (line 158) but not for the previously described MKK2835 x NHP1337 cross. Was this done on different occasions or is it from the same experimental setup? This needs to be clarified.

This is the same experimental set-up. We have added text on lines 176-177 clarifying this and adding the appropriate citation. "A bulk analysis of the uncloned progeny from the MKK2835 × NHP1337 cross was previously published²⁴". The citations were also added above on line 167-168 when the crosses are first introduced.

3. I have a hard time reconciling the Supplementary Table 1 and the description in the text. There are 6 replicates for NF54xNHP4026 in the table but only 3 have been used for infections. Please also indicate in the table which cage has been used. There is an auto-population error for MKK2835xNHP1337. Again, there are 8 cages in the table but only 4 are referenced in the manuscript and only one has been used for the cross.

We have clarified this information and modified the final column in Supplementary Table S1 to indicate which cages were used (or not used) for mosquito infections and the infection method for cages that were used for infection.

4. How are the prevalence and oocyst numbers determined if the sporozoites are used for infection?

We have included additional detail in the methods describing how cages were sampled to determine prevalence and number of oocysts/mosquito (lines 855-856). "For each cage 10 mosquitos were sacrificed on day 10 to measure infection prevalence and number of oocysts/mosquito."

5. Line 183: please define RFLP and MS markers

The unabbreviated names of these technologies are now included in the text.

6. Line 194: How many were selfed vs non-clonal?

This information has been added on line 240 and 246-247.

7. Line 201: What is a cloning round? Different mouse? Different whole infection cycle? Different plate? This does not become clear until the next section: Inbreeding, Outbreeding and Plastid inheritance. And it is not clearly defined. Maybe including it in the schema in Figure 1 would be helpful. Please provide a clear definition before using the term.

We have added further details to aid in clarifying these terms (lines 248-255).

8. Line 230: In the text it says that there were 3 selfed NF54 and no NHP4026 but there are only 2 dots in figure 2A for NF54 and one for NHP4026 in 2B. Maybe I don't understand what data has been included in Figure 2.

We agree that this section is confusing and have modified Figure 2 to add clarity. The old figure showed the parents themselves in gold. We have removed the parental parasites and have included an additional 12 clones genotyped only with microsatellite markers. The legend has been modified to add clarity here. In addition, lines 227-230 and 237-238 have been modified to describe earlier filtering based on MS markers in addition to progeny with full genome sequencing. We also changed the colors of ellipses to match 2C and changed the blue colors in 2A to better differentiate cloning rounds.

Fig 2

9. Line 238: How were the parasites kept for the additional days to 14 and 19 days? Was there a bottle neck (eg most of the parasites taken out for the first round of cloning)? Or was only a small amount removed for cloning and the bulk kept? Was the culture split or was the parasitemia low enough?

We have provided additional details on culturing for each cloning round in Supplemental Table S2. Separate stocks of cryopreserved parasite cultures were thawed for each cloning round that originated from a cryopreserved culture. For each cloning round an aliquot of culture was removed for cloning from a large bulk culture.

10. Line 298: There is not A and B in figure 4.

The legend for Figure 4, and corresponding references to the figure have been corrected.

11. A better description of the strains used here would be useful. Maybe a table with the drug resistance pheno- and genotypes?

Thank you for this great suggestion! We have added Table 1 at line 179 which includes genotype data at known drug resistance loci and phenotype data.

12. Line 313: 3 offspring are CQ resistant which ones? Where are the EC50 values? What is their pfprt status?

Individual progeny CQ IC₅₀ data have been added as panel B in Figure 7, the IC₅₀ QTL scan is now Fig. 7C. We moved the references to CQ resistance phenotypes to later in the paper.

Fig 7

13. Line 325: It has been shown by Carolina Barillas-Mury group that pf47 is important for transmission in the mosquito stages due to evasion of the mosquito immune response (PMCID: PMC5519742, 29229188, PMC5167319). What mosquito strain has been used for this cross?

Anopheles stephensi mosquitos were used for this cross. This information has been added to the methods in line 853-854. We have also added information about this role of pf47 in the discussion at lines 785-786.

14. Line 334: Just to clarify, the bulk analysis published previously and these clonal lines are not from the same cross?

A subset of the cloned progeny described in this paper, and bulk analysis published previously are from the same genetic crossing experiment. We have added text on lines 176-177 to clarify this.

15. Line 337: This is a very vague statement. There is overlap between all of the crosses ever done or just the bulk? Also, it might be helpful to show it in the figure similar to figure 4. In the Sup figure 2 legend it says that there is no

significant overlap. Include the references for the crosses. Is A bulk or clones? Please clarify what you are comparing in the text.

We agree this statement was vague. We removed this whole paragraph from the results as this really isn't adding new results but instead comparing to previously published work and it is already discussed more clearly in the discussion.

16. How is the effective size of a phenotype established? What are some examples from the literature?

The effect size is calculated as phenotypic variation attributable to a genotype at a given locus divided by total variation in phenotype. This definition has been added along with appropriate citations.

Discussion:

17. Lines 466/7: The authors claim that this is the first report of the MKK2835 and NHP1337 cross, yet they published on before. I guess it's the first time they looked at the individual progeny of that cross.

As we state above (response to point 1), while summary data from a subset of one of the three crosses described in this paper has previously been used to validate the inference of selfing seen in bulk segregant approaches, this is the first report on the genetics of individual progeny. We have added a line to results (lines 176-177).

18. Line 471: The high selfing proportion of NHP1337 was observed in the previous paper but was lost *in vitro* over time.

Was there a loss over time for k13 mutants like in the previous bulk population?

The loss of the selfed NHP1337 (the K13 mutant) occurred after day 32 (11 to 15 days of *in vitro* culture) in the bulk analysis. For this cross we cloned on day 2 and day 5 post exsanguination, these two time points are prior to the loss of NHP1337 selfed progeny from the bulk culture. In our cloned progeny, selfed progeny were 54.2% of the positive clones on day 2 and 60.8% on day 5.

19. I am surprised that the sampling over time does not include more of the same genotypes. The authors explain this by under-sampling the possible recombinant progeny. In lines 163/4 it was speculated how many possible new genotypes could have been used to infect the mice used here. Could you use some statistics to back up this claim? You sample about 10% of the possible new progeny. What would be the expected overlap?

We also found this surprising, and thank the reviewer for this comment. To address this a simulation analysis has been added to the supplement as Supplemental Fig S6. We performed simulations for sampling from different

culture sizes and parasitemias using observed oocysts per infected mosquito and the percentages of selfed progeny, and unique recombinants from both crosses. We assumed each unique genotype is represented 1000x to 30000x in the total population of 10,000,000 to 100,000,000 parasites and performed random draws of genotypes. Under this scenario, for 1000 simulated cloning experiments with our sample sizes from each cloning round, we observe 80-90% of experiments have 0 or 1 repeated unique recombinant progeny across cloning rounds and similar numbers of unique recombinant progeny to what we observed. This analysis supports our original inference.

20. Line 532: typo be not by

This has been corrected.

21. Line 535: Please indicate what the expected ratio is.

The expected ratio is 1:1, this was added as a definition in line 335.

22. Line 538: The authors speculate that the selfed progeny will be outcompeted in vitro growth over time. This would agree with their previously published data on the bulk analysis. However, here it looks like MKK2835 is picked up in round 1 and 2 with more clones in round 2 (Fig. S1 B). Although the numbers don't seem to agree with sup table 2 and too many unique recombinants are reported for cloning straight from mouse. It's 18 in the table and a total of 78 but from the figure it looks more like 70 and the total is well above 100.

Thank you for catching this error. The figure was indeed incorrect. A new version has been included. The number of selfed progeny does not change between our cloning rounds at day 2 and day 5 which is in line with the observed results in the bulk analysis where the NHP1337 selfed progeny levels don't change until after day 32 (11-15 days) in *in vitro* culture.

Fig S1

23. It would be interesting to look specifically at the k13 locus on all clones to see if one variant increases over time. As the cloning was done shortly after exsanguination it might not have been enough time *in vitro* for selection. Have the two parental lines ever been co-cultured *in vitro* to detect different growth advantages?

While we agree it would be interesting to look over time at K13 alleles, the experimental set-up does not allow us to draw much inference here. We have added Supplemental figure S2 to explore this. We do see a potential selection on a large region on chromosome 13 but it does not reach the level of genome-wide significance. The two cloning rounds were performed on day 5 and day 2 post exsanguination, at this point we see no selection in the previously published corresponding bulk analysis. We have previously used co-culture experiments to assess the impact of K13 mutations on *in vitro* fitness (Tirrell et al, *Malaria Journal*, 2019 and Nair et al, *Antimicrobial Agents and Chemotherapy* 2018), and agree this is a powerful tool for understanding the fitness costs of specific alleles.

24. When discussing the segregation distortion between the different lines and the genes potentially responsible for this difference it would be useful to actually know the SNPs in these genes and if the isolates are different from each other at this specific locus. The fitness cost for each segment is primarily a function of viability in the mosquito and liver stages as there is only a short amount of *in vitro* growth and no competition with other genotypes due to the cloning. What was the time spread for clones to come up? This could be a useful phenotype to look for genes important in *in vitro* growth.

I'm wondering how much segregation distortion is a function of *in vitro* growth for the highly related parasite lines as well. For the bulk experiment, there was no distortion at the early blood stage but began to manifest after about 2 weeks in culture.

We also find this an intriguing question, though fully addressing the drivers of segregation distortion is beyond the scope of the current manuscript and differences in cloning procedures between cloning rounds make drawing inferences difficult. To understand stage specific selection we have recently performed a bulk analysis across the life-cycle for the NF54 X NHP4026 cross and the data and analysis are now available as a preprint [Kumar et al DOI: 10.1101/2020.09.12.294736]. Interestingly, we see that the segregation distortion on chromosome 7 arises very early during *in vitro* culture, but is not present *in vivo*. The segregation distortion on chromosome 12, and 14 arises much later during *in vitro* culture. The segregation distortion on chromosome 13 is does not reach genome-wide significance in the bulk analysis. A summary of this information has been added in lines 737-741. We have also plotted allele frequencies for each cloning round in Supplemental Fig S2, in this figure it is clear that the segregation distortion on chromosome 7 and 13 is present when

cloning directly from the mouse and the segregation distortion on chromosomes 12 and 14 arises later. This is discussed in lines 369-373 and 733-737.

Fig S2

Figures and Tables:

25. Figure 1. Please don't use a picture of an *Aedes* mosquito. They do not transmit malaria.

We apologize for this fundamental error. We have replaced the image with an image of an anopheline mosquito.

26. Figure 2: I was confused by the different colored cell-like shapes around the parasite dots. Instead of making it clear that they contain clonal parasites they reminded me of hepatocytes. Maybe just using circle around it or joining them by a line? Or maybe have the circle size indicate the number of progenies detected? Also, “* denotes the only observed repeat sampling event between cloning rounds” Shouldn’t it be next to the two dark blue and one light blue dot? Using yellow for the parents is also confusing as it’s not clear in which round it happened. Putting the name next to it will be enough. There is also a dark blue spot in the light blue cluster for MKK2835 cluster. It’s hard to see the difference between round 1 and round 2 as the two blues are very similar. I like the summary of 2C. Maybe these colors could be used in 2A and B to indicate the different occasions? E.g. yellow circles around repeat genotypes within rounds, orange around selfed progeny etc.

Fig. 2 has been modified substantially to address several confusing points (see comment 8 above). We had originally included the actual parents in the figure in gold, the new figure does not include the parents and the ellipses identifying the selfed progeny are orange. The color of the ellipses surrounding each cluster now matches the coloring in Fig. 2C. The shades of blue used for cloning round 1 and 2 were also changed to increase the contrast. Clones that were genotyped only with microsatellite markers have also been added to Figure 2 so that it matches the text.

27. SF1: As the colors for A and B are the same they don’t need to be repeated twice in the legend.

The legend was modified to not repeat the color info.

28. Figure 4: I do not see any black dots in the upper figure even though there is a legend for them. From which cloning round is this data? Is it all combined? Is there a difference between cloning rounds?

The black dots were accidentally left off the figure. This has now been corrected. This figure shows the combined allele frequencies for NF54gfpLuc x NHP4026

from cloning rounds 1, 2 and 3 (red) and NF54WT X NHP4026 cloning round 1 (blue) and all progeny from all cloning rounds for NF54HT-GFP-luc X NHP4026 and NF54WT X NHP4026 (black). The allele frequencies for each cloning round has been included as Supplemental Fig S2 (see comment 24 above).

29. Figure 7B mentions the EC50 values of 35 clones to CQ used for QTL analysis. Please provide the values in a table. Also what is the ES of CQ resistance?

We added an additional panel to Figure 7 (see comment 12 above) providing the IC₅₀ distribution for CQ for this cross (Fig. 7B) and the old panel B is now panel C. We also included additional data so the figure now represents data for 56 progeny. The ES for CQ is provided in the legend (ES=0.84).

30. Sup table 2: How were the numbers for unique recombinant progeny determined for the Thai cross? I don't see how the numbers add up. The days before cloning is confusing for the Thai cross. Why is the cryopreserved source considered 2 days before cloning but round 2? How long was it in culture before sub-cloning?

Our total cloned was initially filtered to remove any clones with low coverage or GQ scores however we neglected to report this filtering in the results section. This has been corrected. The cloning rounds were numbered sequentially, cloning round 1 was initiated on day 5 post transition to *in vitro* culture. Cloning round 2 was initiated several months later from stocks of the bulk progeny that were cryopreserved immediately after transition to *in vitro* culture. A stock was thawed and maintained for 2 days in *in vitro* culture and cloning was initiated after the first reinvasion post thawing.

31. Supplementary cloning protocol. At what day after starting the cloning plate is the PCR done? What is the minimum parasitemia detected? Was there ever a dilution series done to determine sensitivity? I wonder if you only pick-up fast-growing parasites and throw away all with a growth defect that need longer to come up. Also, the empty wells seem to have relatively high CT values as well.

PCR screening was done at weeks 2, 3, 4 and 6. A dilution series was performed and published in Davis et al. Malaria Journal 2020 and our ability to detect low concentrations has been added to the methods (lines 874-876). At week 2 we detect most positives but only expand positives with a CT value of 25 or less into 1 mL cultures. In successive weeks the signals for positive clones with higher CT values become stronger. We have found greater success expanding the clones for cryopreservation and DNA samples if we wait to put them into larger volumes.

Reviewer #2 (Remarks to the Author):

Button-Simons et al. describe generation of two *P. falciparum* genetic crosses in the human liver-chimeric FRG NOD huHep mice. Four *P. falciparum* genetic crosses in chimpanzees have been published, and the authors have also reported a *P. falciparum* genetic cross using the humanized mice previously; therefore, the procedures for generating progeny from *P. falciparum* crosses are not new. The contributions of this report mainly include: 1, generation of relatively large numbers of progeny from two crosses (60 and 84 progeny, respectively); 2, high resolution genetic maps with observations of inheritance bias (or not); 3, estimates of mapping power, which is important for future genetic mapping studies. The manuscript is generally well written, expert discussions are long repeating many results. This study represents an important

contribution to malaria genetic mapping field, including generation of large numbers of progeny.

Some minor points;

32. Abstract: ">140 progeny", better 144 progeny.

This has been updated in the revised manuscript.

33. Line 82-82: Spell out the genes.

Full gene names are now provided.

34. Line 184: the refs are not accurate. The correct refs should be Wellems et al., Nature 1990 345; Su et al., Cell 1997, 91. These are the papers initially used RFLP and microsatellites initially.

We added the Wellems et al. Nature 1990 citation as this was the first report of the HB3xDd2 map and the Su et al. Cell 1997 citation. We retained the Su 1999 citation as this is the first reporting of the full HB3 X Dd2 map based on MS markers. We also added Walliker et al. Science 1987 as it contains RFLP data for the 3D7 X HB3 cross. We retained the Hayton et al. Cell Host & Microbe 2008 to include an initial citation for the 7G8 X GB4 cross.

35. Line 221-225. Can just say the same as A for MKK2835 x NHP1337 cross.

This has been corrected.

36. Line 265, "core region": please define the core regions.

We modified this to core genome to match the terminology used in Miles et al 2016 [doi:10.1101/gr.203711.115] and provided a citation.

37. Line 300: Remove (A) because there is no (B). Or add (B) panel.

This has been corrected in the revised manuscript.

38. Fig. S2. Please provide the source data for the plots of published P. falciparum crosses.

The data used for the HB3 X Dd2, 7G8 X GB4 and 3D7 X HB3 are the sequencing data from Miles et al 2016 [doi:10.1101/gr.203711.115] and the 803 X GB4 data are SNP chip data from Sa et al 2018 [doi:10.1073/pnas.1813386115]. These references have been added to the figure legend.

39. Line 248, "effect size": Is penetrance better? Or functional impact?

In the current context we believe that effect size is the correct term. We have added a definition of effect size to make this clearer.

40. Line 388, “association” and more: For mapping using family data or progeny, linkage is better. Association is more reserved for population based studies.

Here we use association to mean a significant statistical association. We feel this term is more precise than linkage and have modified the text for clarity.

41. Fig. 6, the color lines at the top right corner appear not matching those in the figure.

We have generated a new version of the figure was created with wider lines in the legend to correct this.

42. Line 429, SD: please spell out.

This has been corrected in the revised manuscript.

43. Line 608, “by”: be.

This has been corrected in the revised manuscript.

44. Line 773, Physical recombination map: Suggest removing “physical” to avoid confusion with a physical map.

This has been corrected in the revised manuscript.

45. Line 829: Please provide the approved animal protocol number for the work.

This information has been added to the Declarations section (line 1029-1030).

46. References, some with full titles, some are abbreviated.

This has been corrected in the revised manuscript.

47. The Excels need a title for each table and some explanations. For example, “Days before cloning” in 117502, are the days in culture? Another, 117501, “Post feed exflagellation”, I assume that the numbers are percentages?

This has been corrected in the revised manuscript.

48. The cloning document appears to be an internal protocol; may need some

edits for publication.

The protocol has been revised.

Reviewer #3 (Remarks to the Author):

The manuscript “Surprising variation in two malaria genetic crosses using humanized mice; implications for genetic mapping” by Button-Simons and colleagues is a generally well-written account of the analysis of two genetic crosses between cloned isolates of *Plasmodium falciparum* performed in humanised mice. The major findings of this work involve a power analysis that was performed in order to determine the number of repeats of a particular cross, and the numbers of cloned recombinant progeny that need to be analysed from them in order to detect the genetic drivers of single or multi-genic phenotypes. Additionally, the authors demonstrate that one of their crosses, that between NF54 (or a clone of NF54, presumably) and a culture adapted field isolate from Asia (NHP4026), produced progeny that exhibited significant variation across the genome with respect to the proportions of alleles inherited from one or other of the parents, whilst another of their crosses, this time between two Asian isolates, had a more uniform proportion of parental alleles across the genome. They also found that in both crosses, progeny inherited the maternally-inherited plastid and mitochondrial genomes from both parents, in contrast to previous crosses in *P. falciparum* in which the progeny appeared to have inherited these genomes from only one of the parents presumably due to unequal representation of parental gametocytes during the production of the cross.

Overall, this manuscript is interesting, and represents the results of a huge amount of work, on which the authors are to be commended. I enjoyed reading it, and learnt from it.

49. My major criticism relates to the fact that nowhere in the manuscript is mention made of the previous work performed with rodent malaria parasites which covers very much the same ground; how to deal with maximising the investigative powers of genetic crosses in malaria parasites whilst dealing with uneven representations of parental alleles in progeny which are independent of the phenotype under investigation. I refer to Richard Carter and colleagues “Linkage Group Selection” work, which culminated in Abkallo et al’s PLoS Pathogens paper from 2017 in which the genetic drivers of two independent phenotypes, one of which was multigenic, were investigated using multiple crosses between two parental strains. I completely understand that this was in rodent malaria parasites, but the parallels are obvious, and it seems an odd omission from the current MS.

The lack of acknowledgement of the pivotal rodent malaria work was indeed an oversight. We have added citations of the *P. chabaudi* and *yoelii* LGS work to the introduction where bulk analysis and XQTL were discussed (line 96) and expanded the discussion section (lines 584-593).

50. Indeed, linked to this, throughout the manuscript it was not obvious to me why progeny should be cloned at all. Given that the selection events here should be observable through quantitative-seq LGS (a type of bulk segregation analysis), the extensive time and resources required to clone and genotype individual progeny should be clearly justified. The result given in Figure 4 are the result of producing a cross, cloning large numbers of progeny (months of work), then pooling the individual results from each clone into one analysis. Surely, this could have been done without the need for cloning, and the results would have better resolution for it? This would render the remainder of the MS which deals with power calculations in which the numbers of progeny are a crucial parameter somewhat dispensable.

We agree that bulk approaches are powerful and have invested heavily in developing these tools for *P. falciparum* crosses (i.e. Li, Kumar et al, PLoS Genetic 2019, Kumar et al preprint DOI: 10.1101/2020.09.12.294736). We also believe there are times and applications where individual unique recombinant progeny are necessary. For instance, not all phenotypes are amenable to bulk approaches.

The community resource generated by the chimpanzee genetic crosses has been a major driving force in the quantitative genetics of human malaria. Cloned unique recombinant progeny allow us to determine whether the multiple loci that contribute to a phenotype act additively or epistatically. The ability to identify, and repeatedly phenotype the rare variants at skewed loci is also disproportionately informative for confirming genetic association. Finally, our long-term goal is to perform genetic mapping across complex datatypes, to date there is no technology that will allow simultaneous measurement of phenotype, genome, transcriptome, proteome and metabolome from a single cell.

We agree that Fig. 4 could have been created more efficiently using uncloned progeny. Indeed this figure, where we see large segregation distortion even when cloning immediately after transition to *in vitro* culture led us to design and carry out our previously published work on bulk analysis of uncloned progeny from the MKK2835 X NHP1337 cross.

51. The above criticism is not critical, and can be addressed by inclusion of a discussion of the advantages of cloning over bulk segregation analysis, and specifically q-Seq LGS, in the discussion and possibly the introduction.

We have added citations of the *P. chabaudi* and *yoelii* LGS work to the introduction where bulk analysis and XQTL were discussed (line 96) and expanded the discussion section (lines 584-593) to include a discussion the strengths of traditional cloning based QTL approaches and of bulk selections and of how these methods can complement each other and cited our recent review on this topic Vendrelly Trends in Parasitology 2020.

Specific comments:

52. The Title. Is the variation between crosses really surprising? As the authors point out in the MS, all previous Pf crosses have displayed “segregation distortion”. This is presuming that it is the differences in “segregation distortion” that the authors are referring to as being surprising (!). I think a more suitable title should be used, perhaps one which is more descriptive? Something along the lines of “Genetic mapping of Plasmodium falciparum crosses generated in humanised mice”, but perhaps less boring.

We were surprised that our cross between two recent south east Asian parasites did not display segregation distortion, however, in light of results in other systems this was not actually surprising. We agree that a descriptive title would be more informative and have modified the title accordingly.

53. Results, Lines 163 through 170. The figures quoted here for the maximum number of recombinants from each cross confused me at first, until I realised that they were calculated based on perfect outcrossing, with no selfing. This is highly unrealistic, as in a cross where equal numbers of gametocytes of each parental are provided for the cross, then only 50% of the oocysts would produce recombinants. The only way I can think of to avoid selfing at all, is to use parental clones in which one of the gametocyte sexes is removed from each parent (technically possible, and perhaps an interesting future approach...).

These numbers have been updated to reflect the assumption of random mating rather than perfect outcrossing. Based on cloning results from previous crosses it was unclear to us which assumption was appropriate and we expected the crosses to provide further information. We have also added an example calculation on lines 200-201 to make our assumptions clear.

54. Figure 1 – just a comment but 2/3rds of the time line is taken up with cloning and characterising clones – this could be dramatically reduced if analysis is performed on the uncloned recombinant progeny.

This is true, and indeed cloning is a major bottleneck on our efforts. As we outline above, we believe a joint approach which includes both bulk progeny analysis, and cloned progeny analysis provides us with the tools to answer the broadest range of questions.

55. Line 200; It is not obvious here that the “multiple cloning rounds” referred to were sequential, and what time interval separated them. Please include a clearer description here.

We have modified lines 248-253 to make the cloning round information more clear.

56. Figure 2. This is a nice figure, but giving the clusters different colours, especially colours similar to the “cloning rounds” colours is a bit confusing. Could a black dotted line be drawn around “clusters” instead? Again, the timings of the cloning rounds is crucial information to interpret these results and should be made explicit.

Figure 2 has been updated (see comment 8 above). The colors for each cluster in Figure 2A and 2B previously had no meaning, based on this comment and a comment from reviewer 1 we have modified Figure 2 so that the colors of the ellipses indicate information about whether replicate genotypes were from the same cloning round or different cloning rounds or are parental. We have also added the time in culture before cloning to the legend for each subpanel.

57. Line 230: Information on the relative proportions of gametocytes of each of the parental strains used in the production of the crosses should be given, as this has a major bearing on the interpretation of the subsequent data. It is crucial that this information is added to the results, either in a table, or mentioned explicitly in the text.

We recognize that the relative proportion of gametocytes of each parent strain is critical information as is information on relative infectivity of the gametocytes. For these crosses we recorded the post feed exflagellation, prevalence and oocysts per mosquito for the parental single feeds and for each cross cage (this information is presented in Supplemental Table S1). This information provides an indication of the infectiousness of the parental feeds. In future crosses we will also collect samples of the feed culture so that we have relative ratios of gametocytes for each parent strain.

58. Line 244. The fact that in a perfect cross, 50% of progeny will inherit apicoplast and mitochondrial genomes from Parent A, and 50% from Parent B has never been disputed. The authors set up a bit of a straw man argument here; the fact that two previous *P. falciparum* crosses deviated from this expectation (presumably due to the gametocyte ratios used in their production) was always seen as an experimental anomaly, and not evidence that this is the normal state of affairs. I think it's the sentence “We show here that this is not the case” that is the problem; it gives the impression that people believe that only one parent can contribute these genomes to the recombinant progeny population.

We agree that this facet of prior crosses may have been overstated here, and have revised this statement by removing that sentence and de-emphasizing those results in the abstract and discussion.

59. Figure 3. This is a great figure, and almost by itself justifies all the cloning! However, is it possible to split Panel A into two separate panels showing the results from each of the two rounds of crossing for this cross? This would allow the reader to gauge the extent of the similarity between them. The legend of this

figure mentions the differences between the parents with respect to their sensitivities to anti-malaria drugs for the first time in the MS; this should be mentioned earlier in the text. There is no mention in the legend of what the black horizontal lines on allele frequency graphs are meant to represent. If these are cut-off points for “significance” when considering deviation from mendelian expectations, then this should be stated along with an explanation of how they were calculated.

We have modified Figure 3 to split the progeny into separate groups by cloning round and have added supplemental Fig S2 (see comment 24 above) to show allele frequency split by cloning round.

Fig 3

60. Figure 4: I couldn't find any panel labels on the figure embedded in the MS. I'm presuming the whole genome marker graph is panel A. Do the authors have bulk segregation analysis data for this cross at the same points? If so, it would be really interesting to see that data here in comparison to the results presented. Furthermore, as clones were generated from different cloning rounds, sequentially through time, there is the opportunity to show how allele frequencies change through time here; this would illustrate if the selection valleys observed are the result of selection during the blood stage growth of the parasites, or not. Valleys would be expected to get deeper (and peaks higher) through time, if selection was occurring during culture... It would be fantastic to see the data separated like that here. Any peaks and troughs that do not change their

heights/depths through time must have been the result of selection pre-*in vitro* culture, and therefore be the result of mating incompatibilities or selection during the sporogonic cycle... Again, this figure would be much more informative if the recombinant progeny had not been cloned....

We have just completed an experiment with bulk segregant data for the NF54xNHP4026 cross and have made the data and analysis available as a preprint [Kumar et al DOI: 10.1101/2020.09.12.294736]. Interestingly, we see that the segregation distortion on chromosome 7 arises very early during *in vitro* culture, but is not present *in vivo*. The segregation distortion on chromosome 12, and 14 arises much later during *in vitro* culture. The segregation distortion on chromosome 13 does not rise to the level of genome-wide significance in the bulk analysis. We now reference our preprint in the discussion section.

We have added the allele frequencies divided by cloning round for both the NF54xNHP4026 cross and the MKK2835xNHP1337 cross as Supplemental Fig S2 (see comment 24 above). For the NF54xNHP4026 cross, cloning round date is confounded with cloning in serum vs. albumax and antibiotic treatment regimes to deal with bacterial contamination, so any change in allele frequency cannot be unambiguously tied to time in culture. However, we do see similar patterns to the bulk analysis with the chromosome 7 peak present on day 0 of cloning and the chromosome 12 and 14 peaks present in later cloning rounds. Unlike the bulk analysis the chromosome 13 peak is present when cloning right from the mouse.

61. The coloured panels at the bottom of the figure are not very useful. Instead of Pf_XXXX numbers, is it possible to give the actual names of the genes (where known)? Otherwise the panels are not really informative. Also, on the version of the figure I saw, they are way too small to read properly.

Could the genes under apparent selection be put in a table in the main text?

The colored regions have been modified to include common names where available (see comment 28 above). The genes under selection have been added as a supplemental table (Table S5).

62. Line 320; What does “single genetic variants” mean here?

We agree the use of this term is confusing, we have modified the manuscript to clarify that these are SNPs and microindels.

63. Figure 5: the power analysis is interesting, but again, is it possible to include a model that does not involve cloning?

We agree that a power analysis of bulk segregant analysis is a worthwhile endeavor but beyond the scope of this manuscript. There are many variables that affect this including parental skews in cross progeny, number of recombinants in

the progeny pool used for bulk selection, strength of selection, etc., and including all of these adequately in a power analysis of bulk approaches is better tackled as a separate endeavor and paper. This specific manuscript was designed to describe the cloned progeny from two new genetic crosses and determine the power of traditional mapping with individual clones. We have previously published on a bulk analysis of uncloned progeny for the MKK2835xNHP1337 cross (Li et al PLoS Genetics 2019) and have made the recently completed bulk analysis of a repeat the NF54xNHP4026 cross available as a preprint Kumar et al DOI: 10.1101/2020.09.12.294736. We offer an in depth comparison of benefits of BSA and traditional mapping in Vendrely et al Trends in Parasitology 2020.

64. Line 382. There are many phenotypes that are controlled by single genes in malaria parasites (“genetic traits” is not a good way to describe “phenotype” in this sentence). Could the authors provide references for multi-gene controlled phenotypes in malaria parasites?

We agree there are many phenotypes in malaria where a single genes controls a large proportion of the total variation in the phenotype, some of which have additional loci that further contribute to phenotypic variation. However, whether plasmodium crosses have the necessary power to detect loci with small effect sizes is an open question. We have changed the wording slightly in lines 496-497 and have added appropriate citations. We discuss this in detail in the next paragraph (lines 514-522).

65. Line 400; Again, these results should be discussed in comparison to LGS, in which multiple genotypes may be analysed in a single (albeit repeated) cross, even if the trait is multigenic.

We have modified the discussion section (lines 688-698) to discuss the benefits of LGS for detecting linkage between multiple loci and phenotypes of interest.

66. Lines 438. “Segregation Distortion” I feel the authors may not be considering that the stages of the malaria parasite immediately following fertilisation and up to the appearance of blood stage parasites constitute a long window of potential selection. How do we know that the “segregation distortion” observed in blood stage parasites is not simply the result of selection at the ookinete, oocyst, sporozoite and/or liver stages? Each of these points constitutes a potential bottleneck at which particular genotypes might be favoured over others. This should be considered, and discussed in the MS. Again, it may be possible to separate selection at the mosquito stages from selection at the blood stages by looking for allele frequency movement through time during blood stage culture.

We agree that segregation distortion could arise through selection at any point post meiosis in the complex plasmodium life-cycle. The fact that we see segregation distortion even when we clone directly from the mouse led us to

hypothesize that these skewed allele frequencies might arise through selection in the mosquito stages and led to the study of bulk allele frequency changes in MKK2835xNHP1337 cross. We have added supplemental Fig S2 to show how allele frequencies change for each cloning round. We have also added information to the discussion (lines 737-741) for our recent preprint [Kumar et al DOI: 10.1101/2020.09.12.294736] on a replicate of the NF54 x NHP4026 cross.

67. Line 574: typo – ratio instead of ‘rations’

This has been corrected in the revised manuscript.

68. Line 575: Malaria parasites show adaptation to local vectors. The parasites of S. America are easily transmitted by S. American mosquitoes, and not so well by African/Asian mosquitoes. This could account for differences in allele frequencies observed in crosses between isolates from different continents.

Indeed this is a plausible explanation for segregation distortion that is present when cloning occurred directly after exsanguination as occurred in the NF54xNHP4026 cross. We have added this to the discussion (lines 782-788). We summarize a recently completed BSA analysis on a repeat of the NF54 X NHP4026 cross that has been published as a preprint [Kumar et al DOI: 10.1101/2020.09.12.294736]. This analysis shows that there is no selection present at mouse exsanguination but the selection on chromosome 7 arises almost immediately and later on chromosomes 12 and 14. The selection on chromosome 13 does not reach levels of genome wide significance.

69. Line 596-597. I reread this sentence several times, but could not understand it. Which is the “derived” allele? Could this section be re-written for clarity?

We have rewritten this sentence to improve clarity.

70. Line 618: Again, the possibility of selection at the sporogonic and liver-stages should be considered.

The discussion in lines 782-788 has been expanded to consider selection during other stages.

71. Line 662 Perhaps the word “that” has been accidentally inserted?

This has been corrected in the revised manuscript.

REVIEWERS' COMMENTS:

Reviewer #1 (Remarks to the Author):

Dear authors,

I would like to thank you for incorporating my many suggestions for the paper and reworking the manuscript and figures substantially. I have only a few minor comments and I'm looking forward to see this work in press.

Line 213 and following: This is a bit confusing. Maybe like this: In these crosses diverse/distinct/different F1 progeny are initially present in the blood of the infected FRG NOD huHep/huRBC mouse and must be isolated by limiting dilution cloning to obtain single progeny clones. The isolated progeny may not be clonal due to random distribution of parasites around 0.3 parasites per well. Or something like you had before.

Thanks for including the rational of how parasite lines were filtered. There might be a typo: 168 parasites cloned, then MS filtering => 156 parasites for WGS. 138 recombinants were identified after removing 8 with low genome coverage. That should be $156-8=148$, no? Not 138.

Figure 2 has improved substantially. A minor thing: as the recombinants between cloning rounds have their own color they don't need an asterisk on top of that. In B could you provide the actual number for selfed NHP1337 progeny instead of "many"?

Figure 4 still has no letters but there is a reference to 4C line 417

Figure 7. Maybe mention that you are showing chloroquine IC50 in the legend (B) and maybe also by labelling the y axis with mean CQ IC50.

Table 1: Thank you for generating this table. What does the asterisk mean? What does this CRT mutation do? Is it's not piperaquine related or just not in the Ross data set? Why is the K13 mutation not reported like the others (C580Y and then the AA)? Also, the gene names should all be capital letters as you report the AA change and not the nucleotide. What is eRRSA precisely? This will have to be added to the methods section to make it more understandable.

Table S1: The description makes this is much clearer than before. However, it would help to also label what the cages mean. I assume if it says NF54, mosquitoes were fed on only NF54. This is confusing as it has both parasites lines in the cross tab. Either using just the one strain in the cross column or explaining what pooled or single stains mean would help.

Wrong reference on line 454: should be Table 2 instead of 1

Line 933: reference should be to table S6 not S5. Same in line 940.

Probably best to double check all of the figure and table references now that there are new supplementary figures.

Lines 670-701: It is stated that the parental lines both selfed successfully but it's not clear if they would make it through the liver as they were not used for infections (table S1). Also, the oocyst numbers are much lower than for the other two strains. Could that have an effect as well? Probably not... However, this might actually be worthwhile to mention as you don't need super high mosquito infections for successful mouse infectious further strengthening usefulness and success of the model.

Lines 708-709: NF54 giving very high infection prevalence... Looking at table S1 NF54 had 45% prevalence and 9.2 oocyst per midgut. These are the lowest number for this cross. I understand that infections rates vary greatly between experiments and as NF54 is used regularly I'm sure you do get robust infections generally but the data here do not support this statement. So maybe use robust instead of very high.

Reviewer #3 (Remarks to the Author):

The authors have addressed the reviewers' concerns adequately.

REVIEWERS' COMMENTS:

Reviewer #1 (Remarks to the Author):

Dear authors,

I would like to thank you for incorporating my many suggestions for the paper and reworking the manuscript and figures substantially. I have only a few minor comments and I'm looking forward to see this work in press.

1. Line 213 and following: This is a bit confusing. Maybe like this: In these crosses diverse/distinct/different F1 progeny are initially present in the blood of the infected FRG NOD huHep/huRBC mouse and must be isolated by limiting dilution cloning to obtain single progeny clones. The isolated progeny may not be clonal due to random distribution of parasites around 0.3 parasites per well. Or something like you had before.

Thank you for the suggestion. We have edited this section for clarity and brevity. It now reads "The F₁ progeny in the blood of the infected FRG NOD huHep/huRBC mouse must be cloned by limiting dilution after exsanguination. Isolated progeny may be non-clonal because a small subset of wells were initiated with more than 1 parasite per well."

2. Thanks for including the rational of how parasite lines were filtered. There might be a typo: 168 parasites cloned, then MS filtering => 156 parasites for WGS. 138 recombinants were identified after removing 8 with low genome coverage. That should be $156-8=148$, no? Not 138.

Sorry for the confusion on this. We had forgotten we had resequenced a few samples that had low coverage and had initially included technical replicates in our first round of sequencing. An excel file has been added to github that links the raw data IDs to the clone IDs including the clone IDs of those with only MS data. Our final numbers of samples for this cross are 175 total positive clones, 14 of which were filtered by MS and 161 have full sequencing data. Of the 175, 7 were excluded for low coverage, 25 were non-clonal and 3 were selfed resulting in 140 recombinants of which 84 were unique. This has been corrected in the manuscript (lines 195-198, Figure 2, Supplemental Table 2, and Supplemental Figure 2).

3. Figure 2 has improved substantially. A minor thing: as the recombinants between cloning rounds have their own color they don't need an asterisk on top of that. In B could you provide the actual number for selfed NHP1337 progeny instead of "many"?

The asterisk has been removed and the actual number of selfed NHP1337 progeny (144) is now stated in the legend line 739.

4. Figure 4 still has no letters but there is a reference to 4C line 417

This reference has been corrected and the other references for Figure 4 were checked.

5. Figure 7. Maybe mention that you are showing chloroquine IC50 in the legend (B) and maybe also by labelling the y axis with mean CQ IC50.

This has been corrected.

6. Table 1: Thank you for generating this table. What does the asterisk mean? What does this CRT mutation do? Is it's not piperazine related or just not in the Ross data set? Why is the K13 mutation not reported like the others (C580Y and then the AA)? Also, the gene names should all be capital letters as you report the AA change and not the nucleotide. What is eRRSA precisely? This will have to be added to the methods section to make it more understandable.

The I356T CRT mutation is actually associated with artemisinin resistance not piperazine. This has been corrected in the table. For K13 mutation we wanted to make the point that NHP4026 has no coding mutations in K13 but has a slow clearance phenotype. We have changed this to show the C580Y mutation in the table and have added the information in the legend that NHP4026 has no other coding mutations in K13.

7. Table S1: The description makes this is much clearer than before. However, it would help to also label what the cages mean. I assume if it says NF54, mosquitoes were fed on only NF54. This is confusing as it has both parasites lines in the cross tab. Either using just the one strain in the cross column or explaining what pooled or single stains mean would help.

Definitions for cross and cage have been added and the cage column now specifies that the single parent feeds are controls.

8. Wrong reference on line 454: should be Table 2 instead of 1

This has been corrected.

9. Line 933: reference should be to table S6 not S5. Same in line 940.

This has been corrected.

10. Probably best to double check all of the figure and table references now that there are new supplementary figures.

Supplemental Tables 1 and 2 and all Supplemental Figures were moved to Supplemental File 1 and all references to figures and tables and the new Supplemental Data Files were checked.

11. Lines 670-701: It is stated that the parental lines both selfed successfully but it's not clear if they would make it through the liver as they were not used for infections (table S1). Also, the oocyst numbers are much lower than for the other two strains. Could that have an effect as well? Probably not... However, this might actually be worthwhile to mention as you don't need super high mosquito infections for successful mouse infectious further strengthening usefulness and success of the model.

We actually have data on how well all the infected mice transitioned for this cross including the parental infections. For the NF54WTxNHP4026 infection we infected a mouse via IV injection from the cage of mosquitoes fed on only the NHP4026 parent and via MB from the cage of mosquitoes fed on only on the NF54 parent. We recovered parasitemic blood from both the single parent feeds, NF54 at 0.1% and NHP4026 at 0.05%. These were higher than the parasitemia at time of exsanguination from the mice infected using mosquitoes from the pooled NF54xNHP4026 at 0.013%, 0.017% and 0.02%. This information has been added to Supplemental Table 1 and to the text at lines 170-173 and 397-400. But yes, this data is useful in showing that we can get robust infections even with low oocysts per mosquito.

12. Lines 708-709: NF54 giving very high infection prevalence... Looking at table S1 NF54 had 45% prevalence and 9.2 oocyst per midgut. These are the lowest number for this cross. I understand that infections rates vary greatly between experiments and as NF54 is used regularly I'm sure you do get robust infections generally but the data here do not support this statement. So maybe use robust instead of very high.

We changed the statement as suggested. It now reads "In contrast we observed very few selfed progeny in our allopatric NF54 × NHP4026 cross. Both NF54 and NHP4026 readily infected mosquitoes and had higher parasitemias at time of mouse exsanguination when used alone to inoculate mosquitos than when pooled, with NF54 typically giving robust infections (Supplemental Table 1)."

Reviewer #3 (Remarks to the Author):

The authors have addressed the reviewers' concerns adequately.